



# A Flexible Snow Model (FSM 2.1.0) including a forest canopy

Richard Essery[1], Giulia Mazzotti[2], Sarah Barr[2,3], Tobias Jonas[2], Tristan Quaife[4], and Nick Rutter[5]

[1]School of GeoSciences, University of Edinburgh, Edinburgh, UK
[2]WSL Institute for Snow and Avalanche Research SLF, Davos Dorf, Switzerland
[3]Department of Earth and Environmental Sciences, University of Manchester, Manchester, UK
[4]National Centre for Earth Observation, University of Reading, Reading, UK
[5]Department of Geography and Environmental Sciences, Northumbria University, Newcastle upon Tyne, UK

**Correspondence:** Richard Essery (richard.essery@ed.ac.uk)

**Abstract.** Multiple options for representing physical processes in forest canopies are added to a model with multiple options for representing physical processes in snow on the ground. The canopy processes represented are shortwave and longwave radiative transfer, turbulent transfers of heat and moisture, and interception, sublimation, unloading and melt of snow in the canopy. There are options for Beer's Law or two-stream approximation canopy radiative transfer, linear or non-linear canopy snow interception efficiency, and time/melt-dependent or temperature/wind-dependent canopy snow unloading. Canopy mass and energy balance equations can be solved with one or two model layers. Model behaviour on stand scales is compared with observations of above and below canopy shortwave and longwave radiation, below canopy wind speed, snow mass on the ground and subjective estimates of canopy snow load. Large-scale simulations of snow cover extent, snow mass and albedo for the Northern Hemisphere are compared with observations and land-only simulations by state-of-the-art Earth System Models. Without accounting for uncertainty in forest structure metrics and parameter values, the ranges of multi-physics ensemble simulations are not as wide as seen in intercomparisons of existing models.

## 1 Introduction

Several snow models based on physical principles of energy and mass conservation but offering a range of alternative parametrizations for uncertain energy and mass exchange processes have been developed recently for cryospheric and hydrological applications (Clark et al., 2015; Lafaysse et al., 2017; Niu et al., 2011; Sauter et al., 2020). The Factorial Snow Model (FSM; Essery, 2015) is one such "multi-physics" model that has proved popular with users because the code is freely available (https://github.com/RichardEssery/FSM), compact, easy to use, flexible and thoroughly documented. All of the model state variables and parameters are made accessible through restart and control files for model calibration and data assimilation. At the time of writing, FSM has been used in more than 30 peer-reviewed publications and seven postgraduate theses by users in 11 countries (https://github.com/RichardEssery/FSM/blob/master/publications.md). FSM was originally intended for investigating the range of results from existing physically-based models in simulating snow accumulation and melt on land. Parametrizations of five important processes (decreasing albedo and increasing density of snow with age, increasing thermal conductivity with snow density, storage of liquid water in snow and suppression of turbulent fluxes in stably stratified atmo-



spheric surface layers) can be switched on or off independently, giving up to 32-member ensembles of simulations. Although
neglecting any of these processes is theoretically expected to give poor results, options to neglect them were included because
they are neglected in some existing snow models. In fact, Günther et al. (2020) have subsequently shown that it can be impossi-
ble to distinguish between model configurations that include or neglect specific processes in the face of parameter uncertainty
and with limited evaluation data.

The original version of FSM allowed for partial snow cover on the ground but did not account for exposed vegetation above
snow. Large areas of the Northern Hemisphere have both forests and seasonal snow cover, and influences of forest-snow inter-
actions on weather and climate have been of long-standing interest (Chalita and Le Treut, 1994; Thomas and Rowntree, 1992;
Viterbo and Betts, 1999). Forest canopies can intercept falling snow, shade underlying snow surfaces from solar radiation but
increase incoming thermal radiation, increase drag on the atmosphere and reduce turbulent exchanges between underlying snow
and the air, drop litter that decreases the albedo of underlying snow, and mask the albedo of snow-covered land. Intercepted
snow may unload to the ground, melt in the canopy or sublimate back to the atmosphere, but Lundquist et al. (2021) found
large differences between results from common parametrizations of these processes that are based on limited observations.
Qu and Hall (2014) found large differences in snow-albedo feedbacks for global climate models with differing representations
of snow albedo masking by forests. Climate models have to use simple parametrizations for computational efficiency; they
often represent forests as a single homogeneous layer or simply modify parameters describing the aerodynamic and radiative
properties of the land surface according to vegetation cover. Because a single model layer cannot represent vertical temperature
gradients and decoupling between solar radiation absorption in the upper canopy and thermal emission from the lower canopy,
there has been recent interest in canopy models using two layers (Gouttevin et al., 2015; Todt et al., 2018) or more (Ryder et
al., 2016; Bonan et al., 2018). This is a revival of much earlier work on canopy modelling (e.g., Kondo and Watanabe, 1992;
Yamazaki et al., 1992) that did not immediately translate to global climate models, which still commonly use multi-layer soil
and snow models but single layer ("big leaf") canopy models (Bonan et al., 2021).

This paper describes and demonstrates options added to FSM for representing interactions between snow and forest canopies.
To retain the acronym but to emphasize flexibility over the capability for factorial experiment designs with every possible
combination of options, this version 2 has been renamed as the Flexible Snow Model (FSM2). The multi-layer snow model
used by both FSM and FSM2 is described in Essery (2015). Although many snow models exist, Essery et al. (2012) noted that
a few different process parametrizations are used time and again in different combinations in many of these models; the same
can be said of forest canopy models (Lundquist et al., 2021). As in FSM, FSM2 allows switching between parametrization
options to improve understanding of how they operate together in a complete energy and mass balance snow model. FSM2
can be run with forest canopies represented by one or two model layers and simple or more sophisticated parametrizations of
canopy radiative transfer, snow interception and unloading. Large-scale climate model land surface schemes with vegetation
canopy representations of less or similar complexity to FSM2 include CLASS (Verseghy et al., 1993), CLM (Lawrence et
al., 2019), the Multi-Energy Balance (MEB) component of ISBA (Boone et al., 2017), MATSIRO (Takata et al., 2003), the
MOSES canopy model (Essery et al., 2003) implemented in JULES (Best et al., 2011), SiB (Sellers et al., 1986) and VISA
(Niu and Yang, 2004), but FSM2 does not represent canopy photosynthesis and evapotranspiration. Distributions of snow in





forests on small scales (< 10 m) have been investigated using SnowPALM (Broxton et al., 2015) and an early release FSM

2.0.1 (Essery, 2019; Mazzotti et al., 2020a). After describing substantial developments in FSM2 since version 2.0.1 in section

2 of this paper, the influences of canopy model options on snow simulations at stand and hemispheric scales are demonstrated

with examples in section 3. Results are discussed in relation to other studies in section 4, and this paper concludes with an

outlook on opportunities for applications and developments of FSM2.

## 2 Model description

### 2.1 Forest and canopy model structure

Bulk forest structure is defined in FSM2 by canopy height $h_c$, canopy base height $h_b$ and effective vegetation area index $\Lambda$

including leaves and stems (models that represent transpiration or vegetation dynamics treat leaves and stems separately, but

FSM2 does not). Vegetation density is assumed to be constant with height between the base and the top of the canopy. For a

two-layer canopy model, the fraction of the canopy in the upper layer is set by parameter $f_\Lambda$ (0.5 by default), so the midpoints

of the upper and lower layers are at heights

$$z_1 = h_b + \left(1 - \frac{f_\Lambda}{2}\right)(h_c - h_b) \tag{1}$$

and

$$z_2 = h_b + \frac{1}{2}(1 - f_\Lambda)(h_c - h_b), \tag{2}$$

and the layers have vegetation area indices $\Lambda_1 = f_\Lambda \Lambda$ and $\Lambda_2 = (1 - f_\Lambda)\Lambda$. Rather than being a physically-meaningful and

75 species-dependent canopy base height, $h_b$ is an effective height (2 m by default) for a transition from exponential to logarithmic

wind speed profiles below the canopy.

The heat capacity of vegetation depends on biomass and water content, but models vary widely in how they determine heat

capacity from canopy characteristics, as illustrated in Fig. 1. CLM and MATSIRO neglect canopy heat capacity. SiB and MEB

have a low default heat capacity for dry vegetation that is greatly increased by including the heat capacity of intercepted water.

MOSES and Gouttevin et al. (2015) have higher heat capacities including separate contributions of leaves and trunks. VISA

has an even higher canopy heat capacity, parameterized as a linear function of combined leaf and stem area indices and the

mass of intercepted water. In FSM2, the snow-free vegetation canopy heat capacity is $C_v = C_\Lambda \Lambda$, with the default parameter

value $C_\Lambda = 3.6 \times 10^4$ J K$^{-1}$ m$^{-2}$ chosen to give similar values to Gouttevin et al. (2015). The heat capacity of intercepted

snow is added for a combined calculation of the energy balance of vegetation and snow in the canopy.

### 2.2 Shortwave radiative transfer

Shortwave radiative transfer in canopies has often been represented as an infinite sum of transmissions and reflections between a

single canopy layer and the ground (e.g., Blyth et al., 1999; Stähli et al., 2009; Tribbeck et al., 2004), but a matrix formulation is





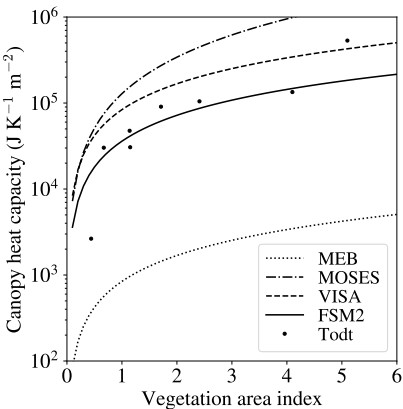

**Figure 1.** Heat capacity of snow-free canopies as functions of vegetation area index in MEB, MOSES, VISA and FSM2. Dots show heat capacities calculated using the method of Gouttevin et al. (2015) with leaf and stem area indices for study sites collated by Todt et al. (2018).

much easier to generalize to multi-layer models (Zhao and Qualls, 2005). Nomenclature for downwards and upwards shortwave radiation fluxes at layer boundaries for one-layer and two-layer canopy models is shown in Fig. 2. Each layer has reflectivities

$R_b$, $R_d$ and transmissivities $\tau_b$, $\tau_d$ for direct-beam and diffuse radiation, respectively, and forward-scattering fraction $s_b$ for direct-beam radiation. Forward scattering and reflections from the canopy and the snow or ground surface (albedo $\alpha$) are assumed to be diffuse. Depending on the availability of measurements for driving FSM2, diffuse and direct-beam shortwave radiation components $S_{\downarrow\text{dif}}$ and $S_{\downarrow\text{dir}}$ above the canopy are read as inputs or global radiation is divided into components using the method of Erbs et al. (1982). The optical properties of canopy layers can either be set by bulk parameters in a Beer's Law

option or calculated from the properties of individual canopy elements in a two-stream approximation. Random orientations are assumed for the canopy elements in either option, although this could be generalized (Otto and Trautmann, 2008).

### 2.2.1   One-layer canopy model

Upwards and downwards diffuse shortwave radiation fluxes at the top and bottom of a single canopy layer are related by

$$S_{\downarrow 1} = R_d S_{\uparrow 1} + \tau_d S_{\downarrow\text{dif}} + s_b S_{\downarrow\text{dir}} \tag{3}$$

$$S_{\uparrow 1} = \alpha S_{\downarrow 1} + \alpha \tau_b S_{\downarrow\text{dir}} \tag{4}$$

$$S_{\uparrow 0} = \tau_d S_{\uparrow 1} + R_d S_{\downarrow\text{dif}} + R_b S_{\downarrow\text{dir}} \tag{5}$$

which can be written as a matrix equation

$$\begin{pmatrix} 1 & -R_d & 0 \\ -\alpha & 1 & 0 \\ 0 & -\tau_d & 1 \end{pmatrix} \begin{pmatrix} S_{\downarrow 1} \\ S_{\uparrow 1} \\ S_{\uparrow 0} \end{pmatrix} = \begin{pmatrix} \tau_d \\ 0 \\ R_d \end{pmatrix} S_{\downarrow\text{dif}} + \begin{pmatrix} s_b \\ \alpha \tau_b \\ R_b \end{pmatrix} S_{\downarrow\text{dir}}. \tag{6}$$



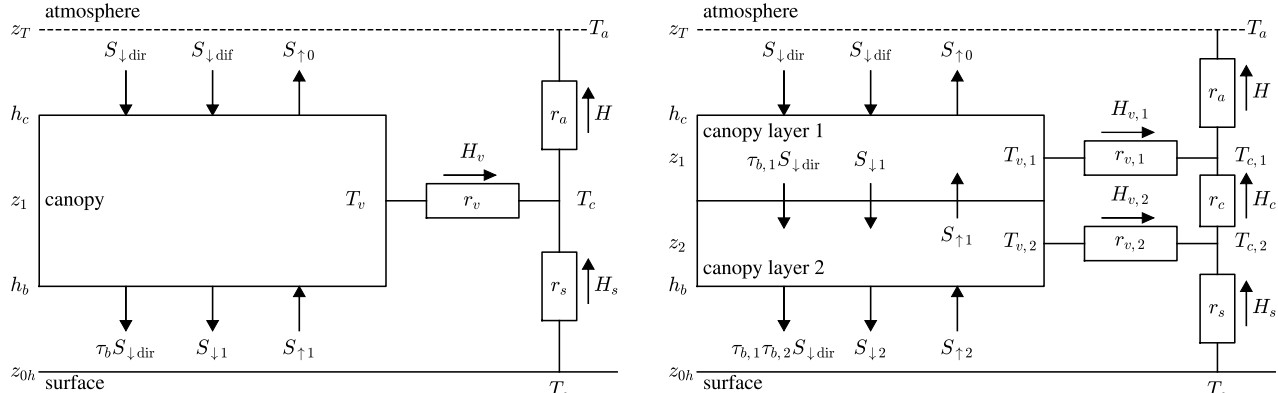

**Figure 2.** Shortwave radiation fluxes, sensible heat fluxes, temperatures and aerodynamic resistances in one-layer (left) and two-layer (right) canopy models. Arrows show the directions in which fluxes are defined to be positive. Longwave radiation and moisture fluxes are numbered and directed in the same ways as shortwave radiation and sensible heat fluxes, respectively.

With fluxes obtained by solving this equation, net shortwave radiation

$$SW_v = S_{\downarrow\mathrm{dif}} - S_{\uparrow 1} + S_{\uparrow 1} - S_{\uparrow 0} + (1 - \tau_b)S_{\downarrow\mathrm{dir}} \tag{7}$$

is absorbed by the vegetation and

$$SW_s = (1 - \alpha)(S_{\downarrow 1} + \tau_b S_{\downarrow\mathrm{dir}}) \tag{8}$$

is absorbed by the snow or ground surface.

### 2.2.2 Two-layer canopy model

The matrix equation for diffuse shortwave radiation fluxes at the boundaries of two canopy layers is

$$
\begin{pmatrix}
1 & 0 & 0 & -R_{d,1} & 0 \\
-\tau_{d,2} & 1 & -R_{d,2} & 0 & 0 \\
0 & -\alpha & 1 & 0 & 0 \\
-R_{d,2} & 0 & -\tau_{d,2} & 1 & 0 \\
0 & 0 & 0 & -\tau_{d,1} & 1
\end{pmatrix}
\begin{pmatrix}
S_{\downarrow 1} \\
S_{\downarrow 2} \\
S_{\uparrow 2} \\
S_{\uparrow 1} \\
S_{\uparrow 0}
\end{pmatrix}
=
\begin{pmatrix}
\tau_{d,1} \\
0 \\
0 \\
0 \\
R_{d,1}
\end{pmatrix}
S_{\downarrow\mathrm{dif}}
+
\begin{pmatrix}
s_{b,1} \\
s_{b,2}\tau_{b,1} \\
\alpha\tau_{b,1}\tau_{b,2} \\
R_{b,2}\tau_{b,1} \\
R_{b,1}
\end{pmatrix}
S_{\downarrow\mathrm{dir}}.
\tag{9}
$$

Net shortwave radiation absorbed by vegetation in the two layers and by the snow or ground surface is

$$SW_{v,1} = S_{\downarrow\mathrm{dif}} - S_{\downarrow 1} + S_{\uparrow 1} - S_{\uparrow 0} + (1 - \tau_{b,1})S_{\downarrow\mathrm{dir}}, \tag{10}$$

$$SW_{v,2} = S_{\downarrow 1} - S_{\downarrow 2} + S_{\uparrow 2} - S_{\uparrow 1} + \tau_{b,1}(1 - \tau_{b,2})S_{\downarrow\mathrm{dir}} \tag{11}$$



and

$$SW_s = (1 - \alpha)(S_{\downarrow 2} + \tau_{b,1}\tau_{b,2}S_{\downarrow\text{dir}}). \tag{12}$$

### 2.2.3 Beer's Law option

The fraction of radiation incident from above at elevation angle $\theta$ transmitted without interception through canopy layer $n$ is parametrized as

$$\tau_{b,n} = \exp(-k_{\text{ext}}\Lambda_n/\sin\theta) \tag{13}$$

with extinction coefficient $k_{\text{ext}} = 0.5$ by default for randomly oriented canopy elements. Integrating Eq. (13) over the sky hemisphere to find the transmission of diffuse radiation through a layer without interception results in an exponential integral (Nijssen and Lettenmaier, 1999) that can be closely approximated by

$$\tau_{d,n} = \exp(-1.6k_{\text{ext}}\Lambda_n). \tag{14}$$

Transmission is thus higher for diffuse than direct-beam radiation for elevation angles less than $39°$. Forward scattering is neglected ($s_{b,n} = 0$). Diffuse and direct-beam canopy layer reflectivities are taken to be $R_{d,n} = (1 - \tau_{d,n})\alpha_c$ and $R_{b,n} = (1 - \tau_{b,n})\alpha_c$ for dense-canopy albedo $\alpha_c$. For a canopy layer with snow cover fraction $f_{cs}$, this is given by

$$\alpha_c = (1 - f_{cs})\alpha_{c0} + f_{cs}\alpha_{cs} \tag{15}$$

for snow-free and snow-covered dense-canopy albedo parameters $\alpha_{c0}$ and $\alpha_{cs}$ (0.1 and 0.4 by default).

### 2.2.4 Two-stream approximation option

From a review of two-stream radiative transfer approximations by Meador and Weaver (1980), Dickinson (1983) adapted the hemispheric constant method for isotropic multiple scattering of light in a homogeneous canopy layer. This was used by Sellers et al. (1986) in SiB and is now used in CLM.

The two-stream model can account for transmission of light through leaves, but FSM2 assumes that canopy elements are opaque and have reflectivities $\alpha_{\Lambda 0}$ when snow free and $\alpha_{\Lambda s}$ when fully covered with snow. Partial canopy snow cover gives

$$\alpha_\Lambda = (1 - f_{cs})\alpha_{\Lambda 0} + f_{cs}\alpha_{\Lambda s}. \tag{16}$$

Solutions of the two-stream equations from Meador and Weaver (1980) involve coefficients

$$\gamma_1 = 2[1 - (1 - \beta)\omega], \ \gamma_2 = 2\beta\omega, \ \gamma_3 = \beta_0, \ \gamma_4 = 1 - \beta_0. \tag{17}$$

For flat, opaque and randomly oriented leaves, $\omega = a_\Lambda$ is the fraction of incident radiation that is scattered, $\beta = 2/3$ is the fraction of scattered diffuse radiation that is directed back into the upward hemisphere and

$$\beta_0 = (0.5 + \mu)\left[1 - \mu\ln\left(\frac{1 + \mu}{\mu}\right)\right] \tag{18}$$





is the upscatter fraction for direct-beam radiation with $\mu = \sin\theta$. The reflectivity and transmissivity of layer $n$ for diffuse radiation are

$$R_{d,n} = \frac{\gamma_2(1 - e^{-2kl})}{k + \gamma_1 + (k - \gamma_1)e^{-2kl}} \tag{19}$$

and

$$\tau_{d,n} = \frac{2ke^{-kl}}{k + \gamma_1 + (k - \gamma_1)e^{-2kl}} \tag{20}$$

for extinction coefficient $k = (\gamma_1^2 - \gamma_2^2)^{1/2}$ and optical thickness $l = k_{\mathrm{ext}}\Lambda_n$. The direct-beam reflectivity, forward-scattering fraction and transmissivity are

$$R_{b,n} = \frac{\omega[(1 - k\mu)(\alpha_2 + k\gamma_3)e^{kl} - (1 + k\mu)(\alpha_2 - k\gamma_3)e^{-kl} - 2k(\gamma_3 - \alpha_2\mu)e^{-l/\mu}]}{(1 - k^2\mu^2)[(k + \gamma_1)e^{kl} + (k - \gamma_1)e^{-kl}]}, \tag{21}$$

$$s_{b,n} = \frac{\omega e^{-l/\mu}[(1 - k\mu)(\alpha_1 - k\gamma_4)e^{-kl} - (1 + k\mu)(\alpha_1 + k\gamma_4)e^{kl}] + 2k\omega(\gamma_4 + \alpha_1\mu)}{(1 - k^2\mu^2)[(k + \gamma_1)e^{kl} + (k - \gamma_1)e^{-kl}]} \tag{22}$$

and

$$\tau_{b,n} = e^{-l/\mu}, \tag{23}$$

where $\alpha_1 = \gamma_1\gamma_4 + \gamma_2\gamma_3$ and $\alpha_2 = \gamma_1\gamma_3 + \gamma_2\gamma_4$.

The albedos of snow and vegetation differ strongly between visible and near-infrared wavelengths. Spectrally resolved short-wave radiation fluxes are available to land surface schemes when coupled to atmospheric models, but they are rarely available from measurements. FSM2 currently only calculates canopy reflectivities and transmissivities from average broadband albedos, but averages of separate visible and near-infrared calculations differ by less than 0.03 for all combinations of $f_{cs}$, $\theta$ and $\Lambda$.

Equation (19) gives the dense-canopy albedo for diffuse radiation as

$$\alpha_c = \frac{\gamma_2}{k + \gamma_1}, \tag{24}$$

which is used to select default parameter values $\alpha_{\Lambda 0} = 0.27$, $\alpha_{\Lambda s} = 0.77$ that match the default snow-free and snow-covered dense-canopy albedos used with Beer's Law. Because of absorption of multiply reflected light, the canopy albedo is much lower than the reflectivity of individual canopy elements even when they are covered with snow. Canopy transmission, calculated by dividing above-canopy by sub-canopy radiation, is higher than canopy layer transmissivities because of downwards reflections. Figure 3 compares canopy albedos and transmission calculated using Beer's Law and the two-stream approximation. The most obvious difference is the systematically greater direct-beam transmission calculated with the two-stream approximation. Transmission of diffuse radiation through a snow-free canopy is slightly lower for the two-stream approximation but can be higher when the canopy is snow covered. Most of the scatter in Fig. 3 comes from differing fractions of canopy snow cover.





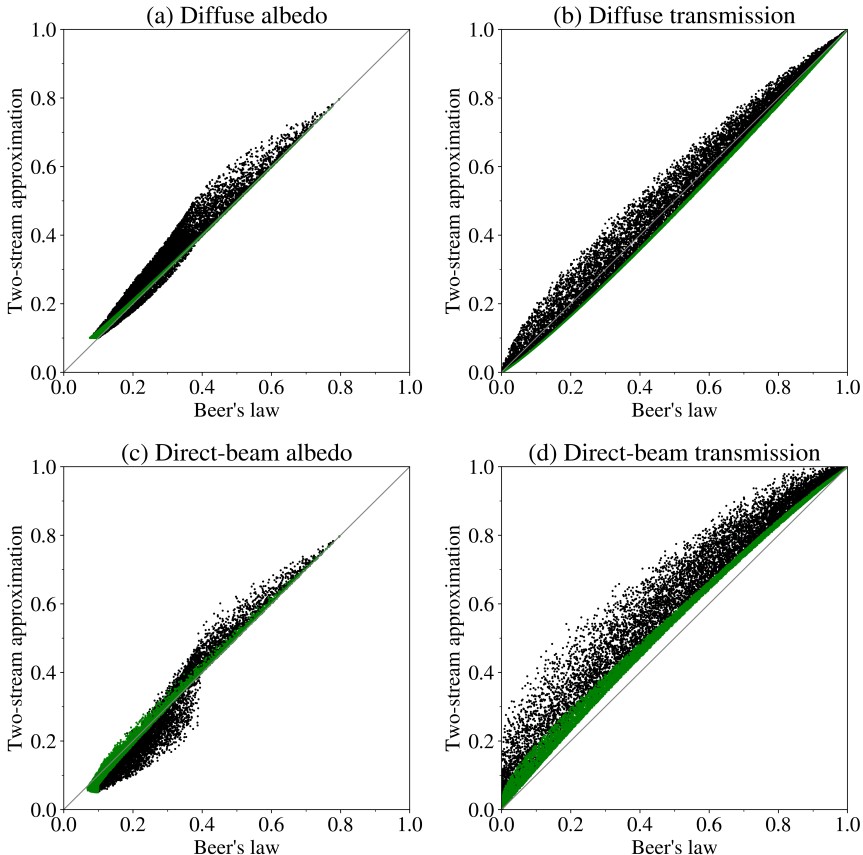

**Figure 3.** Canopy albedo and transmission of diffuse and direct-beam shortwave radiation calculated using Beer's Law and the two-stream approximation in 10,000 simulations with randomly selected values of $\alpha$ (0.1 - 0.8), $\theta$ (5 - 85°), $f_{cs}$ (0 - 1) and $\Lambda$ (0 - 10). Green points are from simulations with snow-free canopies.

## 2.3 Longwave radiation

Vegetation layer $n$ and snow or ground surface temperatures are $T_{v,n}$ and $T_s$. Transmission of longwave radiation from the atmosphere through canopy gaps is given by Eq. 14, in the same way as transmission of diffuse solar radiation. Vegetation, snow and ground emissivities are assumed to be equal to 1, so no longwave radiation is reflected; this makes the radiative transfer equations tractable without resort to matrix solutions.





### 2.3.1 One-layer canopy model

Upwards and downwards longwave radiation fluxes at the top and bottom of a single canopy layer are related by

$$L_{\downarrow 1} = \tau_d LW_\downarrow + (1 - \tau_d)\sigma T_v^4, \tag{25}$$

$$L_{\uparrow 1} = \sigma T_s^4 \tag{26}$$

and

$$L_{\uparrow 0} = \tau_d L_{\uparrow 1} + (1 - \tau_d)\sigma T_v^4, \tag{27}$$

for incoming longwave radiation $LW_\downarrow$ above the canopy and Stefan-Boltzmann constant $\sigma$, from which net longwave radiation

$$LW_v = (1 - \tau_d)(LW_\downarrow - 2\sigma T_v^4 + \sigma T_s^4) \tag{28}$$

is absorbed by the vegetation and

$$LW_s = \tau_d LW_\downarrow + (1 - \tau_d)\sigma T_v^4 - \sigma T_s^4 \tag{29}$$

is absorbed by the snow or ground surface. Upwelling longwave radiation above the canopy is

$$L_{\uparrow 0} = (1 - \tau_d)\sigma T_v^4 + \tau_d \sigma T_s^4. \tag{30}$$

### 2.3.2 Two-layer canopy model

The longwave radiation absorbed by vegetation in the canopy layers and by the surface in a two-layer model are

$$LW_{v,1} = (1 - \tau_{d,1})\left[LW_\downarrow - 2\sigma T_{v,1}^4 + (1 - \tau_{d,2})\sigma T_{v,2}^4 + \tau_{d,2}\sigma T_s^4\right], \tag{31}$$

$$LW_{v,2} = (1 - \tau_{d,2})\left[\tau_{d,1}LW_\downarrow + (1 - \tau_{d,1})\sigma T_{v,1}^4 - 2\sigma T_{v,2}^4 + \sigma T_s^4\right] \tag{32}$$

and

$$LW_s = \tau_{d,1}\tau_{d,2}LW_\downarrow + (1 - \tau_{d,1})\tau_{d,2}\sigma T_{v,1}^4 + (1 - \tau_{d,2})\sigma T_{v,2}^4 - \sigma T_s^4. \tag{33}$$

Upwelling longwave radiation above the canopy is

$$L_{\uparrow 0} = (1 - \tau_{d,1})\sigma T_{v,1}^4 + (1 - \tau_{d,2})\tau_{d,1}\sigma T_{v,2}^4 + \tau_{d,1}\tau_{d,2}\sigma T_s^4. \tag{34}$$

### 2.4 Turbulent fluxes

Vertical momentum, sensible heat and moisture fluxes are parametrized in FSM2 and many other models by integrals of first-order flux-gradient relationships

$$\rho u_*^2 = \rho K_m \frac{\partial U}{\partial z}, \tag{35}$$

$$H = -\rho c_p K_H \frac{\partial T}{\partial z} \tag{36}$$





and

$$E = -\rho K_H \frac{\partial q}{\partial z},$$
(37)

where $u_*$ is the friction velocity, $\rho$ and $c_p$ are the density and heat capacity of air, $K_m$ and $K_H$ are eddy diffusivities for momentum and heat or moisture, $U$ is wind speed, $T$ is air temperature and $q$ is specific humidity. In open areas and above forest canopies ($z > h_c$), the eddy diffusivities are given by the Prandtl hypothesis

$$K_{m,H} = ku_*(z-d)\phi_{m,H}^{-1}\left(\frac{z-d}{L}\right)$$
(38)

for von Kármán constant $k$, displacement height $d$, Obukhov length $L$ and similarity functions $\phi_m$, $\phi_H$ described later.

### 2.4.1  Open areas

Momentum roughness lengths $z_{0f}$ for snow-free ground and $z_{0s}$ for snow on fraction $f_s$ of the ground are combined to give a composite surface roughness length

$$z_0 = z_{0f}^{1-f_s} z_{0s}^{f_s}.$$
(39)

The roughness length for heat transfer from the surface is $z_{0h} = 0.1z_0$, and $d = 0$ in Eq. (38). Integrating Eq. (35) between $z_0$ (where $U = 0$) and wind measurement height $z_U$ (where $U = U_a$) in a surface layer with constant $u_*$ gives

$$u_* = kU_a\left[\ln\left(\frac{z_U}{z_0}\right) - \psi_m\left(\frac{z_U}{L}\right) + \psi_m\left(\frac{z_0}{L}\right)\right]^{-1},$$
(40)

where

$$\psi_m\left(\frac{z}{L}\right) = \int\limits_0^{z/L}\left[\frac{1-\phi_m(\zeta)}{\zeta}\right]d\zeta.$$
(41)

Wind speeds at heights $z < z_U$ are given by

$$U(z) = \frac{u_*}{k}\left[\ln\left(\frac{z}{z_0}\right) - \psi_m\left(\frac{z}{L}\right) + \psi_m\left(\frac{z_0}{L}\right)\right].$$
(42)

Integrating Eq. (36) between $z_{0h}$ (where $T = T_s$) and temperature measurement height $z_T$ (where $T = T_a$) with constant $H$ gives

$$H = \frac{\rho c_p}{r_a}(T_s - T_a)$$
(43)

for aerodynamic resistance

$$r_a = \frac{1}{ku_*}\left[\ln\left(\frac{z_T}{z_{0h}}\right) - \psi_H\left(\frac{z_T}{L}\right) + \psi_H\left(\frac{z_{0h}}{L}\right)\right]$$
(44)





and

$$\psi_H\left(\frac{z}{L}\right) = \int_0^{z/L} \left[\frac{1 - \phi_H(\zeta)}{\zeta}\right] d\zeta. \tag{45}$$

Moisture flux

$$E = \chi_s \frac{\rho}{r_a}[q_{\text{sat}}(T_s) - q_a] \tag{46}$$

is calculated using the same aerodynamic resistance as the sensible heat flux and a moisture availability factor $\chi_s = 1$ if $E < 0$ (condensation) or if there is snow on the ground, or

$$\chi_s = \frac{r_a}{r_a + r_{sg}} \tag{47}$$

otherwise, where $r_{sg}$ is the resistance for evaporation from soil moisture (a fixed parameter in the absence of interactive soil moisture and photosynthesis models in FSM2). $q_a$ is the specific humidity measured at height $z_T$ and $q_{\text{sat}}(T)$ is the saturation humidity at temperature $T$. Latent heat flux $LE$ is calculated by multiplying the moisture flux by the latent heat of sublimation if $T_s < T_m$ or the latent heat of evaporation otherwise.

### 2.4.2 Forest areas

Aerodynamic resistances and turbulent fluxes above, within and beneath forest canopies are shown for one-layer and two-layer canopy models in Fig. 2. Resistances $r_a$, $r_{v,n}$ and $r_s$ couple the upper canopy air space (temperature $T_{c,1}$) to the atmosphere, vegetation layers (temperatures $T_{v,n}$) to corresponding canopy air space layers, and the ground or snow surface to the lower canopy air space, respectively. An additional resistance $r_c$ couples the upper and lower canopy air space layers in the two-layer model. The sensible heat fluxes are parametrized as

$$H = \frac{\rho c_p}{r_a}(T_{c,1} - T_a) \tag{48}$$

from the upper canopy air space to the atmosphere,

$$H_{v,n} = \frac{\rho c_p}{r_{v,n}}(T_{v,n} - T_{c,n}) \tag{49}$$

from vegetation layer $n$ to canopy air space layer $n$, and

$$H_s = \frac{\rho c_p}{r_s}(T_s - T_{c,N}) \tag{50}$$

from the surface to the lower canopy air space ($N = 1$ for a one-layer canopy model or 2 for a two-layer canopy model). The flux between canopy air space layers in a two-layer model is

$$H_c = \frac{\rho c_p}{r_c}(T_{c,2} - T_{c,1}). \tag{51}$$





The corresponding moisture fluxes are

$$E = \frac{\rho}{r_a}(q_{c,1} - q_a),\tag{52}$$

$$E_{v,n} = \chi_{v,n}\frac{\rho}{r_{v,n}}[q_{\text{sat}}(T_{v,n}) - q_{c,n}],\tag{53}$$

$$E_s = \chi_s\frac{\rho}{r_s}[q_{\text{sat}}(T_s) - q_{c,N}]\tag{54}$$

and

$$E_c = \frac{\rho}{r_c}(q_{c,2} - q_{c,1}),\tag{55}$$

where $q_{c,n}$ is the specific humidity of canopy air space layer $n$. The moisture availability factor for evaporation from vegetation layer $n$ is $\chi_{v,n} = 1$ if $E_{v,n} < 0$ or

$$\chi_{v,n} = f_{cs,n} + (1 - f_{cs,n})\frac{r_{v,n}}{r_{v,n} + r_{sv}}\tag{56}$$

otherwise, where $r_{sv}$ is a fixed resistance for evaporation from snow-free vegetation (100 s m$^{-1}$) by default.

Displacement height $d = 0.67h_c$ and vegetation roughness length $z_{0v} = 0.1h_c$ are used for dense canopies. FSM2 and many other models (e.g., Choudhury and Monteith, 1998; Dolman, 1993; Koivusalo and Kokkonen, 2002; Boone et al., 2017) follow Inoue (1963) in assuming exponential wind profiles within dense canopies. Eddy diffusivities are continuous at the canopy top and have an exponential form

$$K(z) = K(h_c)\exp\left[\eta\left(\frac{z}{h_c} - 1\right)\right],\tag{57}$$

with $\eta = 2.5$ by default. Within-canopy stability effects are neglected. The wind speed profile above, within and below the canopy is

$$U(z) = \begin{cases} \frac{u_*}{k}\left[\ln\left(\frac{z-d}{z_{0v}}\right) - \psi_m\left(\frac{z-d}{L}\right) + \psi_m\left(\frac{z_{0v}}{L}\right)\right] & z \geq h_c \\ U_c\exp\left[\eta\left(\frac{z}{h_c} - 1\right)\right] & h_b < z < h_c \\ U_b\ln\left(\frac{z}{z_0}\right)\left[\ln\left(\frac{h_b}{z_0}\right)\right]^{-1} & z_0 \leq z \leq h_b \end{cases}\tag{58}$$

with

$$u_* = kU_a\left[\ln\left(\frac{z_U - d}{z_{0v}}\right) - \psi_m\left(\frac{z_U - d}{L}\right) + \psi_m\left(\frac{z_{0v}}{L}\right)\right]^{-1}.\tag{59}$$

Continuity is used to calculate wind speeds $U_c$ and $U_b$ at canopy top and base heights $h_c$ and $h_b$. Integrating Eq. (36) between the relevant heights gives aerodynamic resistances

$$r_a = \frac{1}{ku_*}\left[\ln\left(\frac{z_T - d}{h_c - d}\right) - \psi_H\left(\frac{z_T - d}{L}\right) + \psi_H\left(\frac{h_c - d}{L}\right)\right] + \frac{h_c\left[e^{\eta(1 - z_1/h_c)} - 1\right]}{\eta K_H(h_c)}\tag{60}$$





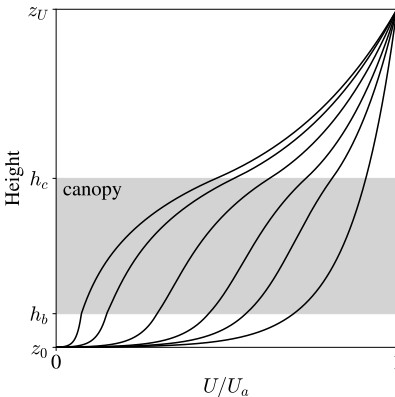

**Figure 4.** Wind speeds above, within and below canopies with vegetation area indices (from right to left) 0, 0.5, 1, 2, 4 and 8.

for heat transfer between the highest canopy layer and the atmosphere,

$$r_c = \frac{e^\eta h_c}{\eta K_H(h_c)} \left( e^{-\eta z_2/h_c} - e^{-\eta z_1/h_c} \right) \tag{61}$$

between heights $z_1$ and $z_2$ within the canopy and

$$r_s = \frac{1}{k^2 U_b} \ln\left(\frac{h_b}{z_0}\right) \ln\left(\frac{h_b}{z_{0h}}\right) + \frac{e^\eta h_c}{\eta K_H(h_c)} \left( e^{-\eta h_b/h_c} - e^{-\eta z_N/h_c} \right) \tag{62}$$

between the surface and the lowest canopy layer. The resistance for sensible heat flux from vegetation to the air within canopy layer $n$ is given by

$$\frac{1}{r_{v,n}} = C_{\text{leaf}} \Lambda_n U(z_n)^{1/2} \tag{63}$$

with $C_{\text{leaf}} = 0.05$ m$^{1/2}$ s$^{-1/2}$ by default. Calculating conductance $r_v^{-1}$ rather than resistance $r_v$ avoids dividing by small numbers as $\Lambda \to 0$. Canopy conductances of this form are stated in several model description papers without reference or simply with reference to earlier models. The $U^{1/2}$ dependence is characteristic of engineering expressions for laminar flow over plates and appears in biophysical literature for flow over leaves at least as far back as Raschke (1960).

Wang (2012) pointed out that exponential wind speed profiles do not satisfy no-slip boundary conditions at the ground and do not converge to logarithmic profiles for zero canopy density. Moreover, Inoue (1963) predicted that $\eta$ should be a function of canopy density, but models generally take it to be a constant parameter. The logarithmic wind speed profile below height $h_b$ in Eq. (58) is commonly adopted in models to impose a no-slip boundary condition. In FSM2, dense-canopy and open conductances are weighted by the vertically projected vegetation fraction $f_v = 1 - \exp(-k_{\text{ext}}\Lambda)$ and $(1 - f_v)$, respectively, and combined in parallel to get resistances for sparse canopies. Sub-canopy wind speeds are calculated as weighted averages of wind speeds from Eqs (42) and (58), as shown in Fig. 4.





### 2.4.3 Stability functions

Atmospheric stability is characterized by Obukhov length

$$L = -\frac{\rho c_p T_a u_*^3}{kgH}. \tag{64}$$

As in Bonan et al. (2018) (but without modification for a roughness sublayer above canopies), the stability functions used in FSM2 are

$$\phi_m(\zeta) = \begin{cases} (1 - 16\zeta)^{-1/4} & \zeta < 0 \\ 1 + 5\zeta & \zeta \geq 0 \end{cases} \tag{65}$$

and

$$\phi_H(\zeta) = \begin{cases} (1 - 16\zeta)^{-1/2} & \zeta < 0 \\ 1 + 5\zeta & \zeta \geq 0 \end{cases} \tag{66}$$

with $\zeta$ limited to the range -2 to 1. These functions integrate to give

$$\psi_m(\zeta) = \begin{cases} 2\ln\left(\frac{1+x}{2}\right) + \ln\left(\frac{1+x^2}{2}\right) - 2\tan^{-1}x + \frac{\pi}{2} & \zeta < 0 \\ -5\zeta & \zeta \geq 0 \end{cases} \tag{67}$$

and

$$\psi_H(\zeta) = \begin{cases} 2\ln\left(\frac{1+x^2}{2}\right) & \zeta < 0 \\ -5\zeta & \zeta \geq 0, \end{cases} \tag{68}$$

where $x = (1 - 16\zeta)^{1/4}$. The Obukhov length depends on fluxes, the fluxes depend on the stability functions and the stability functions depend on the Obukhov length, so stability adjustments have to be calculated iteratively.

## 2.5 Conducted heat fluxes

Snow and soil temperatures and liquid water fractions are simulated by a multi-layer heat conduction model (Essery, 2015). The conducted heat flux into the snow or ground surface is calculated as

$$G = \frac{2\lambda_1}{\Delta z_1}(T_s - T_1) \tag{69}$$

where $\lambda_1$ is the thermal conductivity of snow or soil and $T_1$ is the temperature of a surface layer of thickness $\Delta z_1$ (0.1 m by default).





## 2.6 Energy balance

### 2.6.1 Open areas

The surface energy balance

$$f(T_s) = LW_s + SW_s - G - H - LE - L_f M = 0 \tag{70}$$

with latent heat of fusion $L_f$ and parametrizations for the fluxes is a nonlinear equation for the unknown surface temperature and snowmelt rate $M$. The solution is first found with $M = 0$. From an initial guess of temperature $T_{s0}$ and neglecting the complicated temperature dependence of $r_a$ if stability adjustments are applied, a linear estimate of $T_s$ is given by

$$T_s = T_{s0} - f(T_{s0}) \left( \frac{df}{dT_s} \right)^{-1} = T_{s0} + f(T_{s0}) \left[ 4\sigma T_{s0}^3 + \frac{2\lambda_1}{\Delta z_1} + \frac{\rho}{r_a}(c_p + LD_s \psi_s) \right]^{-1}, \tag{71}$$

where

$$D_s = \left. \frac{dq_{\text{sat}}}{dT} \right|_{T=T_{s0}} = \frac{Lq_{\text{sat}}(T_{s0})}{R_{\text{wat}} T_{s0}^2} \tag{72}$$

and $R_{\text{wat}}$ is the gas constant for water vapour. A single evaluation of Eq. (71) gives an approximate solution, and repeated evaluations with $T_s$ calculated in each iteration being used as $T_{s0}$ in the next is the Newton-Raphson method for solving Eq.

(70). If this gives $T_s > T_m$ and there is snow with ice mass $I$ on the ground, iteration of Eq. (71) is repeated assuming that all of the snow melts and $M = I/\delta t$. If this gives $T_s < T_m$, the snow does not all melt and $T_s = T_m$ is known; Equation (70) is then solved instead for the unknown melt rate by substitution of $T_s = T_m$ in the equations for the other fluxes.

### 2.6.2 Forest areas

For a one-layer canopy model, energy and water vapour mass conservation equations

$$f_1 = LW_s + SW_s - G - H_s - LE_s - L_f M = 0, \tag{73}$$

$$f_2 = LW_v + SW_v - H_v - LE_v - C_v \frac{dT_v}{dt} = 0, \tag{74}$$

$$f_3 = (H - H_s - H_v)/(\rho c_p) = 0, \tag{75}$$

and

$$f_4 = (E - E_s - E_v)/\rho = 0 \tag{76}$$

with parametrizations for the fluxes form a set of four nonlinear equations with four unknowns: $q_c$, $T_c$, $T_v$ and either $T_s$ or $M$. Writing vectors $\boldsymbol{f} = (f_1, f_2, f_3, f_4)^T$ and $\boldsymbol{x} = (T_s, q_c, T_c, T_v)^T$, a solution without melt is first found by iterating

$$\boldsymbol{x} = \boldsymbol{x_0} - \mathrm{J}^{-1} \boldsymbol{f}(\boldsymbol{x_0}) \tag{77}$$





where J is the Jacobian matrix of $\boldsymbol{f}$ with elements

$$J_{ij} = \frac{\partial f_i}{\partial x_j} \tag{78}$$

given in the full documentation distributed with the FSM2 code. Equation (77) is implemented by solving

$$\mathrm{J}(\boldsymbol{x} - \boldsymbol{x_0}) = -\boldsymbol{f}(\boldsymbol{x_0}) \tag{79}$$

numerically and iterating to find $\boldsymbol{x}$. If the solution has $T_s > T_m$ and there is snow with ice mass $I$ on the ground, Eq. (79) is iterated again with $M = I/\delta t$, assuming that all of the snow melts in the timestep. If this gives $T_s < T_m$, the snow does not all melt; the surface temperature is then known and Eq. (79) is solved to find the unknown melt rate.

There are seven energy and water vapour mass conservation equations for the surface and two canopy layers, and seven unknowns in a vector $\boldsymbol{x} = (T_s, q_{c,1}, T_{c,1}, T_{v,1}, q_{c,2}, T_{c,2}, T_{v,2})^T$. The conservation equations and the elements of the 7×7 Jacobian matrix are, again, listed in the FSM2 documentation. Solutions are found in the same way as for the one-layer canopy.

## 2.7  Canopy snow

Early land surface models used the same interception capacities for liquid water and snow held on vegetation (e.g., the canopy capacity per unit VAI was 0.2 kg m$^{-2}$ in CLASS prior to version 3.1 and 0.1 kg m$^{-2}$ in CLM prior to version 5.0), but measured canopy snow loads can actually be much higher. Subsequently, many models have adopted the representation of snow interception and unloading developed from observations by Hedstrom and Pomeroy (1998). Lundquist et al. (2021) noted that snow interception in global models is still based on a few geographically limited observations.

    In FSM2, a forest canopy intercepts a fraction of falling snow up to a maximum $S_c = S_\Lambda \Lambda$, with $S_\Lambda = 4.4$ kg m$^{-2}$ by default
(Essery et al., 2003). The fraction of the canopy covered with snow, which is required for canopy albedo and sublimation calculations, is parametrized as

$$f_{cs} = \left(\frac{S_v}{S_c}\right)^{2/3} \tag{80}$$

for a canopy layer with intercepted snow mass $S_v$. The 2/3 exponent here is quoted in many papers without citation or explanation. In fact, it was introduced by Deardorff (1978), who proposed it as a compromise between values of 0 and 1 that
would make evaporation of dew from vegetation too fast and too slow, respectively.

    The mass balance equation for snow in a canopy layer with interception rate $I_v$, sublimation rate $E_v$, melt rate $M_v$ and unloading rate $U_v$ is

$$\frac{dS_v}{dt} = I_v - E_v - M_v - U_v. \tag{81}$$

If vegetation temperature $T_{v0}$ calculated without melt exceeds $T_m$ while intercepted snow remains, the canopy snowmelt rate
is

$$M_v = \min\left[\frac{C_v}{L_f \delta t}(T_{v0} - T_m), \frac{S_v}{\delta t}\right] \tag{82}$$





and the vegetation temperature is reset to

$$T_v = T_{v0} - \frac{L_f M_v \delta t}{C_v}. \tag{83}$$

Snowfall that is not intercepted is added to the snow on the ground with the same fresh snow density as for snow falling in
open areas, but snow unloading from the canopy is added with the same density as snow already on the ground. Meltwater
dripping from the canopy and rain are added to the snow as liquid water. Interception, unloading and melt are calculated for
both layers in the two-layer canopy model; the lower layer can intercept throughfall of snow but not unloading or drip from the
upper layer.

### 2.7.1 Canopy interception options

CLM and MATSIRO intercept constant fractions of snowfall until the canopy capacity is reached. This linear option is imple-
mented in FSM2 as

$$I_v = \min\left[f_v S_f, \frac{(S_c - S_v)}{\delta t}\right]. \tag{84}$$

CLASS, ISBA, JULES and VISA use the Hedstrom and Pomeroy (1998) interception rate model

$$I_v = \frac{(S_c - S_v)}{\delta t}\left[1 - \exp\left(-\frac{f_v S_f \delta t}{S_c}\right)\right], \tag{85}$$

which gives the same interception rate as Eq. (84) for an initially snow-free canopy but approaches $S_c$ more slowly. Because
of the non-linearity in Eq. (85), one-layer and two-layer representations of the same canopy density have different interception
efficiencies.

### 2.7.2 Canopy unloading options

Canopy snow loads decrease exponentially with time after snowfall in CLASS, JULES and MEB. This time-dependent option
is implemented in FSM2 along with increased unloading when canopy snow is melting as

$$U_v = \frac{S_v}{\tau_u} + m_u M_v, \tag{86}$$

with $\tau_u = 10$ days (Bartlett and Verseghy, 2015) and $m_u = 0.4$ by default (Storck et al., 2002).

CLM5 and VISA have unloading rates that depend on canopy temperature and wind speed. This temperature/wind-dependent
unloading option is implemented in FSM2 as

$$U_v = \left[\frac{1}{c_T}\max(T_v - 270.15, 0) + \frac{U_a}{c_U}\right]S_v, \tag{87}$$

with $c_T = 1.87 \times 10^5$ K s and $c_U = 1.56 \times 10^5$ m by default (Roesch et al., 2001). With the default parameter values, unloading
rates from Eq. 87 exceed rates from Eq. 86 whenever the canopy temperature is above -2.8°C or the wind speed is above 0.2
m s$^{-1}$.



## 3 Model test results

The following comparisons of FSM2 results with observations are presented as demonstrations of model behaviour rather than rigorous evaluations of the model. Simulations of sub-canopy shortwave radiation, longwave radiation and wind speeds are first compared with observations at sites in Finland, Sweden and Switzerland. A complete simulation of snow mass and energy balance is then compared with a year of observations at a site in Switzerland. Finally, FSM2 simulations are compared with observations and Land-Surface, Snow and Soil moisture Model Intercomparison Project (LS3MIP) simulations of Northern Hemisphere snow cover extent, snow mass and albedo.

### 3.1 Sub-canopy radiation

Radiation was measured with arrays of ten shortwave radiometers and four longwave radiometers under sparse forest canopies over periods of between 4 and 23 days during March and April 2011 at Abisko, Sweden (68.3°N, 18.8°E) and during March and April 2012 at Sodankylä, Finland (67.4°N, 26.6°E). Measurements were made in five stands described by Reid et al. (2013) at each site: leafless birch stands with average sky view fractions determined from hemispherical photography between 0.59 and 0.9 at Abisko, and pine, spruce and mixed stands with average sky view fractions between 0.41 and 0.72 at Sodankylä. There was snow on the ground but the canopy was snow free during all of the measurement periods. For consistency in the model, vegetation area indices were obtained from sky view fractions by inverting Eq. (14). Above-canopy meteorological driving datasets were constructed using measurements from the Abisko Scientific Research Station and the Finnish Meteorological Institute Arctic Space Centre at Sodankylä.

Figure 5 shows average sub-canopy shortwave radiation and shortwave radiation transmission calculated as average sub-canopy radiation divided by average above-canopy radiation over the measurement periods. Simulations are shown with Beer's Law and two-stream radiative transfer options; the matrix formulation of the radiative transfer equations ensures that one- and two-layer canopy models give the same results if the albedos of the layers are the same. As was seen in Fig. 3, diffuse transmission is slightly higher and direct-beam transmission is lower for snow-free canopies if Beer's Law is used. Averaged over the measurement periods for each of the stands, the simulated transmission in Fig. 5 (b) is lower for Beer's Law, particularly for measurement periods during which lower fractions of the cumulated incoming shortwave radiation was diffuse (these fractions were between 34% and 59% over the measurement periods).

In analogy with shortwave radiation transmission, dividing average sub-canopy longwave radiation by average above-canopy longwave radiation gives a canopy longwave enhancement factor. This is generally greater than one because canopy elements have higher emissivities than the atmosphere, particularly when the sky is clear (Rutter et al., 2023). The canopy longwave enhancement emphasizes differences in model canopy temperature rather than differences in above-canopy longwave radiation for the different measurement periods in Fig. 6. The upper canopy layer shades the lower layer from shortwave radiation by day and traps longwave radiation by night in a two-layer canopy model (Gouttevin et al., 2015), but average differences between FSM2 simulations with one and two layers are small for these sparse canopies.



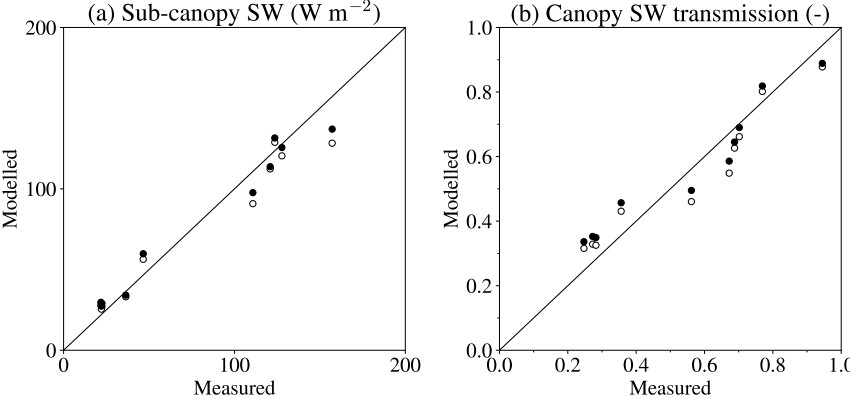

**Figure 5.** Beer's Law (open circles) and two-stream approximation (closed circles) simulations of (a) average sub-canopy downwards short-wave radiation and (b) canopy shortwave transmission, compared with measurements in 10 forest stands. Simulations with one and two canopy layers are indistinguishable.

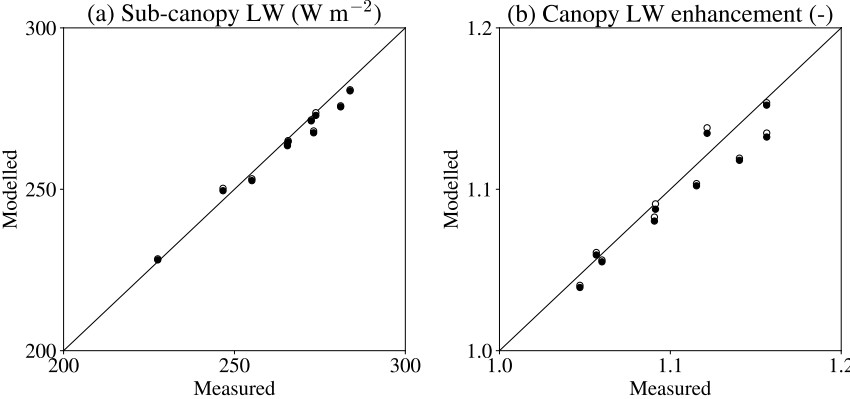

**Figure 6.** Simulations with one canopy layer (open circles) and two canopy layers (closed circles) of (a) average sub-canopy downwards longwave radiation and (b) canopy longwave enhancement, compared with measurements in 10 forest stands.

## 3.2 Sub-canopy wind speed

Wind speed was measured by single 2D sonic anemometers at 2 m height in open areas and under forest canopies in four pine stands (8 February to 5 March 2018) and 14 spruce stands (10 January to 25 April 2018 and 24 January to 13 April 2019) near Davos, Switzerland (46.8°N, 9.9°E) and four pine stands near Sodankylä (17 to 29 April 2019) with Λ ranging from 1 to 3.4 (Mazzotti et al., 2020a). Figure 7 compares simulated forest wind speeds and ratios of forest to open wind speeds averaged over the measurement periods, which varied from 3 to 58 days in length. The observations may be influenced by forest edge effects and there is a large degree of scatter in the simulations, but the simulated ratios have a moderate correlation (0.64) with





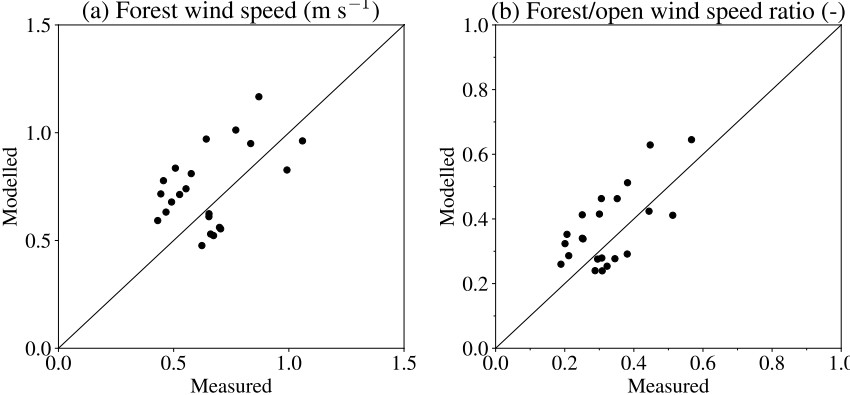

**Figure 7.** Simulations of (a) average sub-canopy wind speeds and (b) ratios between sub-canopy and open wind speeds, compared with measurements in 22 forest stands. Simulations with one and two canopy layers are indistinguishable.

the observations and a moderate root mean square error (0.1). Equation (58) gives a continuous vertical profile of wind speeds without discretisation, so results for one- and two-layer canopy models do not differ.

## 3.3 Snow simulations at a site

A site has to be well characterised, well instrumented and well maintained to provide direct measurements of all of the inputs required by energy balance models. Several such sites have been used in the Snow Model Intercomparison Project (SnowMIP; Etchevers et al., 2004; Essery et al., 2009; Menard et al., 2019) for evaluation of snow models. Forest and open meadow sites at Alptal, Switzerland (Stähli and Gustafsson, 2006; Stähli et al., 2009) that were used in SnowMIP2 were also used by Gouttevin

et al. (2015) and are used again here to demonstrate the performance of FSM2. The forest stand is dominated by spruce and fir with typical heights of 25 m and an average VAI of 3.96. The mild climate can allow snow cover to appear and disappear several times over the winter; such conditions are known to be challenging for snow modelling (Essery et al., 2009). FSM2 was run for the winter of 2004-2005 at Alptal with all driving data except precipitation measured above the forest canopy on a 35 m high mast. Precipitation was measured with a sheltered gauge in the meadow and gauge-corrected by scaling snowfall

to match snow accumulation in the meadow between snow-free conditions observed on 13 December 2004 and the maximum snow mass of 352 $\mathrm{kg\,m^{-2}}$ observed on 14 March 2005. Model parameters were adjusted to match measured snow-free albedos above the forest and the meadow, but all other parameters were left at default values to focus on differences in simulations due to differences in model options. Figure 8 compares simulations with observations of average snow mass measured weekly along 30 m transects in the forest and in the meadow, albedo and outgoing (upward) longwave radiation measured at fixed

points above the forest and in the meadow, shortwave transmission and incoming (downward) longwave radiation measured with radiometers moving along a 10 m horizontal rail under the canopy, and a subjective estimate of canopy snow load made by an observer from 0 (snow-free canopy) to 8 (maximum possible snow interception), scaled to the model's canopy capacity. The





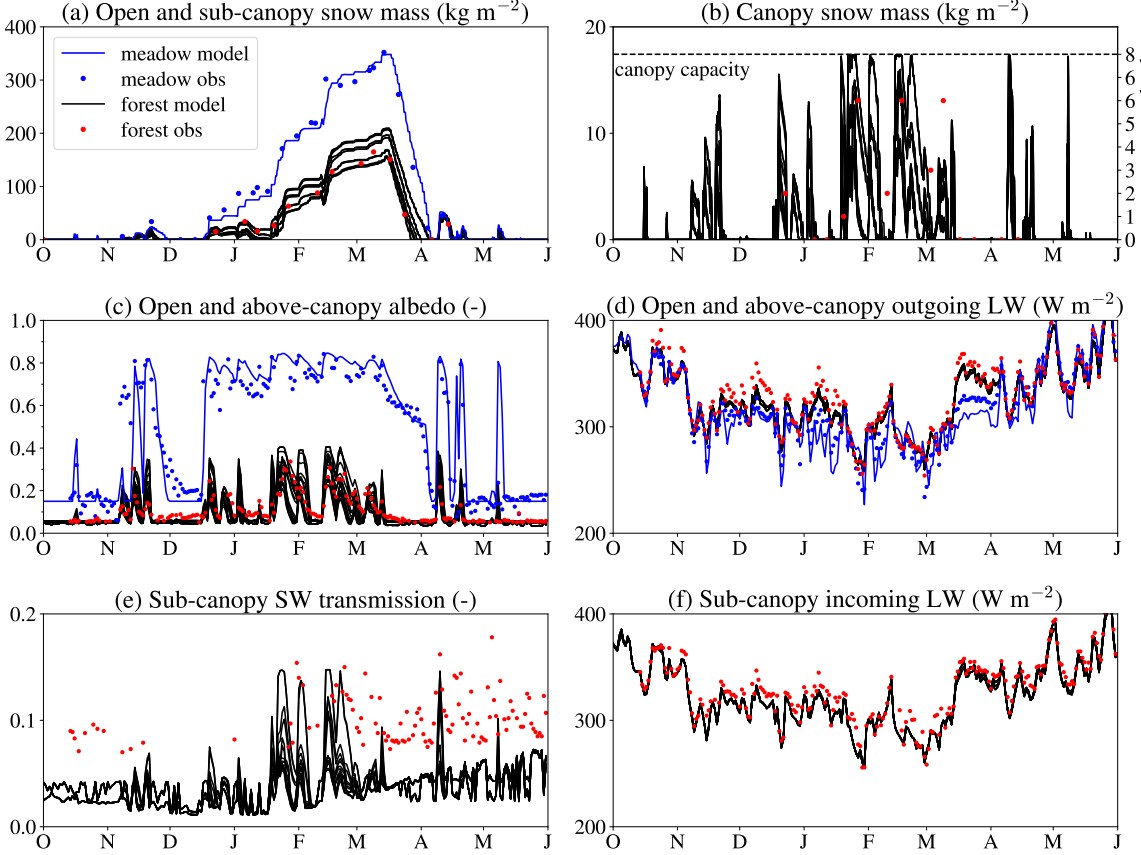

**Figure 8.** Simulations (lines) and observations (points) for the Alptal forest and meadow sites in 2004-2005. Albedo, transmission and longwave radiation fluxes are daily values. Subjective canopy snow load observations are scaled to the model's canopy capacity (dashed line in b).

albedo, transmission and incoming longwave radiation measurements have been filtered to remove periods when it is suspected that there was snow on the upward-facing radiometers. Lower snow mass, lower albedo and higher outgoing longwave radiation for the forest than for the meadow are apparent in both observations and simulations.

The same model options were used for snow on the ground in the meadow and the forest, corresponding to FSM "configuration 31": prognostic snow albedo, variable thermal conductivity, prognostic snow density, stability adjustment of the turbulent exchange coefficient and prognostic liquid water content. Because cumulated snowfall in the model driving data was forced to match the observed maximum snow mass in the meadow, it is not surprising that the simulation matches the snow accumulation well, but the model also matches observed snowmelt well, including short periods with shallow snow cover in November 2004
and April 2005. Only a shallow snow cover is required to increase the meadow albedo from below 0.2 to above 0.7.



There are sixteen simulations for the forest: with linear or nonlinear interception efficiency, with one or two canopy layers, with Beer's Law or two-stream canopy radiative transfer, and with time/melt-dependent or temperature/wind-dependent unloading. The forest simulations have a 56 kg m$^{-2}$ range in maximum snow mass and a 47 day range in duration of snow cover (Fig. 8a and Table 1). Although large compared with plausible observation errors, these ranges are smaller than spread seen in comparisons of forest snow simulations by different models (Rutter et al., 2009, and Appendix A). The forest canopy is dense enough that snow on the ground has very little influence on above-canopy albedo (Fig. 8c). The albedo increases when there is intercepted snow in the canopy but remains below 0.4. Agreement between the durations of periods with elevated albedo in observations and simulations suggests that the simulated persistence of snow in the canopy is realistic. Simulated transmission of shortwave radiation through the canopy (Fig. 8e) increases when there is intercepted snow in the canopy but is generally lower than observed; the transmission could be increased with little impact on simulated snow masses by decreasing the parameter $k_{\mathrm{ext}}$. In the lower layer of a two-layer canopy model, daytime heating by shortwave radiation and nighttime cooling by longwave radiation are reduced by the shelter of the upper layer (Gouttevin et al., 2015; Todt et al., 2018). This reduces the diurnal range in sub-canopy longwave radiation, but differences in simulations of daily-average longwave radiation (Fig. 8d and f) are small.

There are too many lines on Fig. 8a to identify the individual canopy model configurations, so Table 1 gives the maximum sub-canopy snow mass, the duration of snow cover on the ground and the fraction of total snowfall sublimating for each simulation. Variations in maximum snow mass explain a large fraction of the variation in snow cover duration ($r^2 = 0.91$). The unloading option has the largest influence on snow on the ground; snow unloads from the canopy faster with the temperature/wind-dependent unloading option and accumulates on the ground, where it is sheltered from wind and sublimation. Less snowfall is intercepted by the nonlinear interception option as the canopy load increases, so this option also increases the mass of snow on the ground. Differences in transmission of shortwave radiation through the canopy between the Beer's Law and two-stream radiative transfer options depend on canopy snow and sky conditions (Fig. 3 b and d), but these differences have little influence on snow beneath the canopy because the transmission is always low for the dense Alptal forest canopy. The number of model canopy layers has complex influences on snow simulations. The upper layer in a two-layer model intercepts more snowfall than the lower layer, and that snow is exposed to higher wind speeds and shortwave radiation. Snow in the lower layer is sheltered, but the overall effect is for slightly more snow sublimation. The shading of the lower layer, however, decreases daytime sub-canopy longwave radiation, which reduces mid-winter melt and delays the final disappearance of snow on the ground in spring.

### 3.3.1 Sensitvity to canopy density

The amounts of snowfall reaching the ground in open and forest sites differ because some of the intercepted snow in the canopy sublimates, and meltwater dripping from snow in the canopy drains from the snow on the ground if it does not refreeze. Melt rates on the ground differ between sites because the canopy modifies the shortwave radiation, longwave radiation and turbulent heat fluxes in the surface energy balance. All of these differences are influenced by the canopy density, which is represented by vegetation area index in FSM2. Figure 9 explores variations with VAI in simulations using the Alptal 2004-2005 meteorology.



**Table 1.** Maximum sub-canopy snow mass, duration of snow cover on the ground and fraction of total snowfall sublimating in the 16 forest simulations in Fig. 8a with every possible combination of linear or nonlinear snowfall interception, one or two canopy layers, Beer's Law or two-stream canopy radiative transfer, and time/melt-dependent ($t - M$) or temperature/wind-dependent ($T - U$) canopy snow unloading.

| Interception | Layers | Radiation | Unloading | Mass (kg m$^{-2}$) | Duration (days) | Sublimation |
|---|---|---|---|---|---|---|
| linear | one | Beer's Law | $t - M$ | 156 | 75 | 14% |
| linear | one | Beer's Law | $T - U$ | 192 | 108 | 10% |
| linear | one | two-stream | $t - M$ | 154 | 74 | 14% |
| linear | one | two-stream | $T - U$ | 191 | 110 | 10% |
| linear | two | Beer's Law | $t - M$ | 154 | 85 | 16% |
| linear | two | Beer's Law | $T - U$ | 206 | 119 | 11% |
| linear | two | two-stream | $t - M$ | 158 | 85 | 15% |
| linear | two | two-stream | $T - U$ | 208 | 119 | 11% |
| nonlinear | one | Beer's Law | $t - M$ | 169 | 97 | 13% |
| nonlinear | one | Beer's Law | $T - U$ | 197 | 110 | 9% |
| nonlinear | one | two-stream | $t - M$ | 167 | 97 | 13% |
| nonlinear | one | two-stream | $T - U$ | 196 | 112 | 9% |
| nonlinear | two | Beer's Law | $t - M$ | 167 | 100 | 14% |
| nonlinear | two | Beer's Law | $T - U$ | 209 | 120 | 9% |
| nonlinear | two | two-stream | $t - M$ | 168 | 100 | 14% |
| nonlinear | two | two-stream | $T - U$ | 210 | 121 | 10% |

Figure 9a shows fractions of snowfall that sublimate from the ground and canopy combined. Light winds (below 4 m s$^{-1}$ for 97% of hours) give low sublimation for the open meadow site (VAI = 0), but sublimation from the canopy increases as the density increases. Simulations are continuous as VAI $\rightarrow$ 0 by design in FSM2. Differences between simulations increase as canopy density increases and level out for high canopy densities, limited by energy availability. Reference to Table 1 again 495 allows identification of the individual model configurations in Fig. 9a; simulations with the time/melt-dependent unloading option consistently have the highest sublimation because snow remains exposed in the canopy for longer.

The contributions of energy by net shortwave radiation, net longwave radiation and sensible heat fluxes to melt snow on the ground in simulations with varying canopy density are shown in Fig. 9b. This is for simulations with linear interception, one canopy layer, Beer's Law radiative transfer and time/melt-dependent unloading (the first row in Table 1); other canopy options 500 give fluxes that differ in detail but follow the same trends with canopy density. Simulated melt at the Alptal meadow site is dominated by shortwave radiation, with a small contribution from sensible heat fluxes, and net longwave radiation is a small loss of energy for snowmelt. As the canopy density increases and sky view under the canopy decreases, the net shortwave radiation decreases and the net longwave radiation increases to become the dominant source for melt energy under dense canopies. Sensible heat fluxes first increase as canopy air space temperatures increase because of canopy heating and then





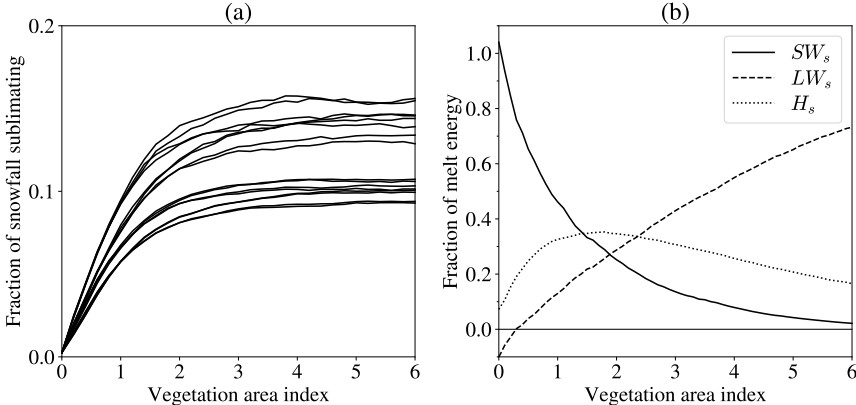

**Figure 9.** Sensitivity to canopy density in simulations with the Alptal 2004-2005 meteorology. (a) Fractions of total snowfall sublimating in simulations with the 16 canopy model configurations. (b) Contributions of net shortwave radiation, net longwave radiation and sensible heat fluxes to energy for melting snow on the ground in simulations with linear interception, one canopy layer, Beer's Law radiative transfer and time/melt-dependent unloading.

decrease as sub-canopy wind speeds decrease with increasing canopy density. Latent and ground heat fluxes (not shown) each contribute less than 10% of the melt energy beneath canopies.

### 3.4  Northern Hemisphere snow simulations

Simulation of snow on a grid covering an area requires coupling with an atmospheric model or distributed driving data that are not directly available from measurements. Meteorological reanalyses can be used for simulations over large areas at coarse

resolutions; for example, Brun et al. (2013) used ERA-Interim reanalyses to drive the Crocus snow model over northern Eurasia. Here, the performance of FSM2 for simulating Northern Hemisphere seasonal snow cover at 0.5° resolution is demonstrated with 2000-2010 driving data from the GSWP3 bias-corrected reanalysis (Kim, 2017), which was previously used in LS3MIP (van den Hurk et al., 2016). Canopy heights, vegetation area indices, forest fractions and snow-free albedos for FSM2 are taken from global maps developed by Lawrence and Chase (2007) for CLM; deciduous and evergreen forest fractions are

shown in Fig. 10. Separate FSM2 simulations with parameters for evergreen forest, deciduous forest and unforested land are combined to give the averaged seasonal cycles of Northern Hemisphere snow area and mass shown in Fig. 11 for the 16 canopy model configurations listed in Table 1. The same model configuration as in section 3.3 was used for unforested land and snow interception was set to zero for leafless deciduous forest canopies, so differences between simulations are dominated by areas with evergreen forests.

Figure 11 compares FSM2 simulations with the nine models that submitted results for LS3MIP and estimates from multi-dataset historical snow extent and snow mass time series (Mudryk et al., 2020). One of the LS3MIP models follows the estimated snow mass closely but underestimates snow extent. FSM2 and the other LS3MIP models give very similar snow



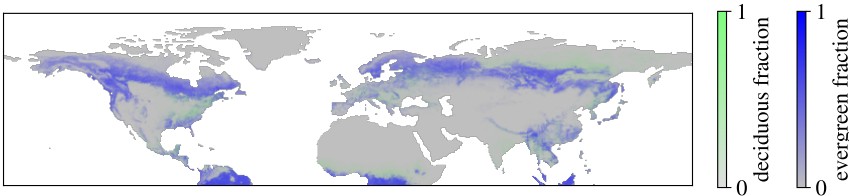

**Figure 10.** Northern Hemisphere deciduous and evergreen forest fractions from Lawrence and Chase (2007).

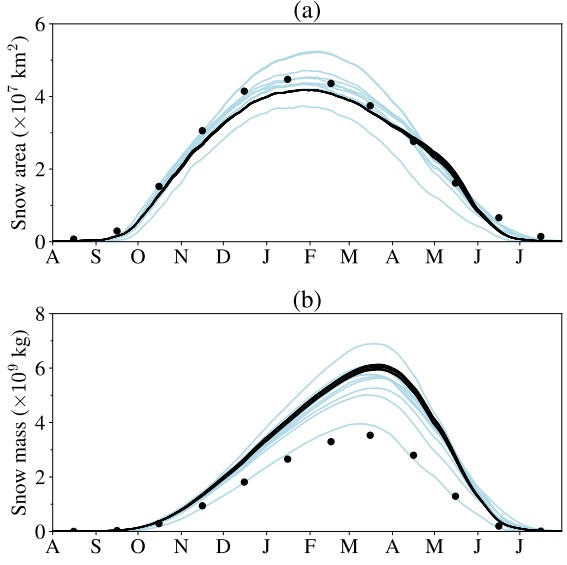

**Figure 11.** Northern Hemisphere seasonal snow area (a) and mass (b) averaged over 2000-2010 from 16 FSM2 simulations (black lines), nine LS3MIP models (blue lines) and estimates from multi-dataset historical time series (circles).

masses while persistent snow cover accumulates at high latitudes through October and November. Thereafter, the models retain different amounts of ephemeral snow mass and spread out. FSM2 tends to overestimate snow mass and underestimate the peak snow area in comparison with the historical timeseries but lies within the ranges of the LS3MIP models. Despite noticeable local differences in snow mass on the ground under forests demonstrated in section 3.3, the spread in hemispheric averages for FSM2 simulations in Figure 11 is small. This is because evergreen neeedleleaf forests only cover 8% of the Northern Hemisphere land area in the dataset of Lawrence and Chase (2007) and around 20% of the area with seasonal snow cover in FSM2 simulations.



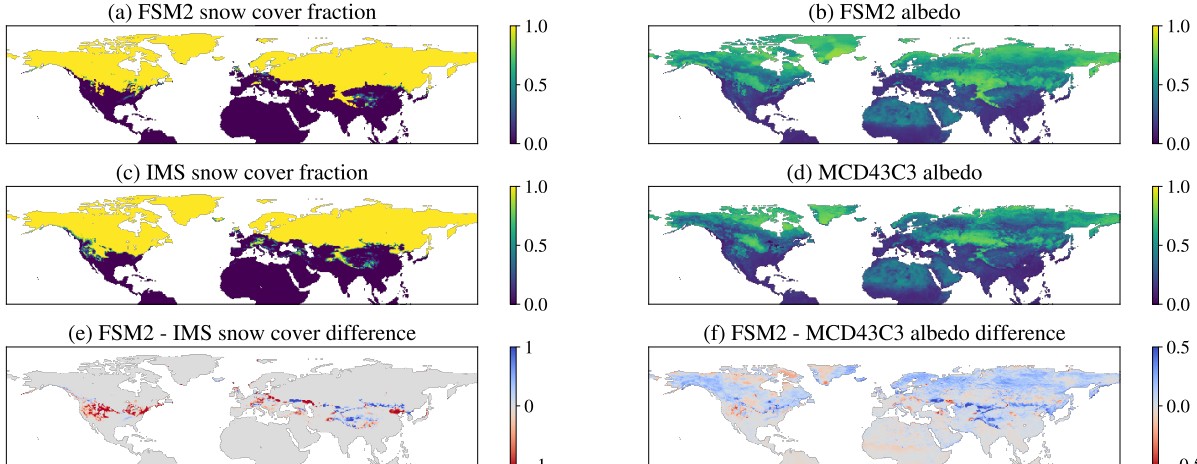

**Figure 12.** Northern Hemisphere snow cover fraction and albedo maps for 1 March 2010 from FSM2, IMS and MODIS MCD43C3. The FSM2 simulation has two canopy layers, two-stream radiative transfer, nonlinear canopy snow interception and temperature/wind-dependent unloading.

As an example of the spatial distribution of Northern Hemisphere snow cover, Fig. 12 compares FSM2 with binary snow cover information at 4 km resolution from the Interactive Multi-sensor Snow and Ice Mapping Service (IMS; US National Ice Center, 2008) and white sky albedo at 0.05° resolution from the MODIS MCD43C3 dataset (Schaaf and Wang, 2021) aggregated to 0.5° resolution on 1 March 2010. The good match in hemispheric snow cover extent at this time of year has some compensation between underestimates and overestimates at the southern limits of snow cover (Figure A3 shows similar snow cover difference maps for the LS3MIP models). Albedo differences are largest where FSM2 has errors in snow cover fraction. FSM2 and MODIS both show dark bands across the continents where the albedo of snow is masked by boreal forests, but the FSM2 albedo is generally higher. Evaluation and optimization of FSM2 for hemispheric snow simulations would clearly require much larger samples of observations and simulations, but these preliminary results are encouraging.

## 4 Discussion

Despite longstanding interest, uncertainty remains in how best to represent canopy and sub-canopy snow processes in models. In contrast with FSM2, Gouttevin et al. (2015) made a number of different design decisions when implementing a canopy model in SNOWPACK. For a two-layer model, SNOWPACK conceptualizes the canopy as having an upper leaf layer and a lower trunk layer, without multiple reflections of shortwave radiation in the upper layer or interception of snow in the lower layer. Considering that turbulent transport within vegetation canopies is still poorly understood, Gouttevin et al. (2015) followed Blyth et al. (1999) in using logarithmic wind profiles even within canopies and making empirical adjustments to surface aerodynamic resistances beneath canopies for simplicity. The two vegetation layers in SNOWPACK are coupled to a



single canopy air space ($T_{c,1} = T_{c,2}$, $r_{v,1} = r_{v,2}$ and $r_c = 0$ in Fig. 2) and a minimum heat exchange coefficient for windless conditions is included in the parametrizations of fluxes between the canopy air space and the atmosphere. The aerodynamic resistance between the canopy and the canopy air space in SNOWPACK (Eq. 32 in Gouttevin et al. (2015)) does not scale with

leaf area and can be more than two orders of magnitude smaller than the same resistance in FSM2 (Eq. 63). Differences in simulations of maximum snow mass on the ground at Alptal in March 2005 between one-layer and two-layer canopy models are smaller in FSM2 (2 to 17 kg m$^{-2}$ in Table 1) than in SNOWPACK (54 kg m$^{-2}$ in Figure 4 of Gouttevin et al. (2015)).

    Unloading of canopy snow does not occur in SNOWPACK until the intercepted snow load exceeds the canopy capcity (Eq. 5 in Gouttevin et al. (2015)), whereas unloading is continuous in FSM2 (Eq. 86 or 87). Rain does not interact with canopy snow in

FSM2, but this has been investigated in SNOWPACK by Bouchard et al. (2024). The canopy snow unloading parametrizations reviewed by Lundquist et al. (2021) gave the largest differences in simulations of snow on the ground at Alptal when coupled in the full mass and energy balances of FSM2. Lundquist et al. (2021) found little experimental evidence for a maximum snow interception capacity in published datasets. In fact, there is also little evidence for a maximum capacity in the Alptal forest simulations because the model's capacity is rarely reached (Fig. 8b).

FSM2 is formulated to limit differences between one-layer and two-layer canopy models for reasons that are purely related to the methods chosen for numerical solution of the mass and energy balance equations. If the albedos and temperatures of both layers in a two-layer model are equal, the sub-canopy shortwave radiation and longwave radiation will be the same as in a one-layer model. The Hedstrom and Pomeroy (1998) snow interception, however, was developed as a bulk canopy model. The nonlinearity of Eq. (85) means that a two-layer canopy model has a slightly higher interception efficiency than a one-layer

model with the same canopy capacity. The vertical distribution of snow in the canopy is important for sublimation because snow higher in the canopy is exposed to higher wind speeds and higher solar radiation. Bonan et al. (2021) argued that five to ten canopy layers are necessary to model turbulent and radiative fluxes but did not consider interception. Terrestrial laser scanning can now be used to make measurements of canopy loading (Russell et al., 2021) that might be useful in developing an explicitly multilayer interception model.

Qu and Hall (2014) reported a large spread between climate models for the albedo of snow-covered land in boreal forest regions, with implications for simulations of snow albedo feedback. Different canopy radiative transfer parametrizations, however, need not result in large differences in masking of snow albedo by forests. FSM2 parameters have been adjusted to give similar canopy albedos from Beer's Law and the two-stream approximation; transmission of shortwave radiation through the canopy differs, but this has little influence on snowmelt beneath dense canopies. Variations in canopy parameters, VAI maps

and alternative canopy structure metrics have not been explored here but can give large variations in canopy albedo (Essery, 2013; Malle et al., 2021).

## 5   Outlook

The addition of canopy model options enables the use of FSM2 for investigating snow-forest interactions and for large-scale simulations including forested regions. During the course of its development, FSM2 has been coupled with the Snow Mi-



crowave Radiative Transfer model (SMRT; Sandells et al., 2017), the Multiscale Snow Data Assimilation System (MuSA; Alonso-González et al., 2022) and the Swiss Operational Snow-hydrological model system (OSHD; Mott et al., 2023). It has been adapted for metre-resolution simulation of snow in discontinuous forests (Mazzotti et al., 2020a, b, 2023). Implementations of FSM2 in OSHD (Quéno et al., 2023) and the Canadian Hydrological Model (CHM; Marsh et al., 2020) add representations of horizontal snow redistribution at snowdrift-resolving scales. FSM2 is now being coupled with the Open

Global Glacier Model (OGGM; Maussion et al., 2019) for modelling snow and firn mass balances on glaciers. Parametrizations for interception of snow by deciduous conifers such as larch and trapping of drifting snow by tundra shrubs would be other useful additions.

In a comparison with observations on small scales at sites with varying climatic conditions, FSM2 captures the differences in snow mass, snow cover duration and albedo between open and forested sites. For Northern Hemisphere seasonal snow area

and mass simulations, FSM2 lies within the range of land surface schemes from state-of-the-art Earth System Models. Because FSM2 is intended for snow research and does not attempt to include all the land processes required in modern Earth System Models, it is more compact and easier to use than these land surface schemes. There are 3,186 lines of code in the FSM2 source directory, compared with more than 190,000 for CLM5.0, and the FSM2 code compiles in under five seconds. The Northern Hemisphere simulations presented here took about seven minutes of CPU time to run 50,411 land points for 2,920 timesteps

per year on a single 1.5 GHz processor. No communication is required between points, so the code is trivially parrallelizable. FSM2 offers opportunities for investigations requiring snow simulations on large grids, in large ensembles or for long periods.

*Code and data availability.* The FSM 2.1.0 code and user documentation are available from https://zenodo.org/records/13308507 (doi: 10.5281/zenodo.13308507, published 12 August 2024) and scripts to produce the figures in this manuscript are available from https://zenodo.org/ records/13863749 (doi:10.5281/zenodo.13863749, published 30 September 2024). Alptal data are distributed with the code to run examples

of forest and meadow snow simulations. The other freely-available datasets used in this work are:

Abisko sub-canopy radiation data, http://catalogue.ceda.ac.uk/uuid/6947880b98d32e249a8638ebe768efd2

Sodankylä sub-canopy radiation data, http://catalogue.ceda.ac.uk/uuid/9c8c86ed78ae4836a336d45cbb6a757c

Davos and Sodankylä sub-canopy wind data, doi:10.16904/envidat.162

GSWP3 forcing data, https://wiki.c2sm.ethz.ch/LS3MIP/GeneralInformationOnLS3MIP

IMS daily Northern Hemisphere snow cover, doi:10.7265/N52R3PMC

LS3MIP model outputs, https://esgf-index1.ceda.ac.uk/search/cmip6-ceda/

MODIS MC43C3 Version 6.1 albedo, doi:10.5067/MODIS/MCD43C3.061

Northern Hemisphere snow extent and mass, doi:10.18164/cc133287-1a07-4588-b3b8-40d714edd90e

*Author contributions.* RE and TJ managed the FSM2 code development. Code was contributed by RE, GM and TQ. GM, SB and NR

contributed field data. This manuscript was prepared by RE and edited by all authors.



*Competing interests.* The authors declare that they have no competing interests

*Acknowledgements.* Development of FSM2 was supported by NERC grants NE/P011926/1 and NE/X005194/1, and Swiss National Science Foundation grants 200021_16921 and P500PN_202741. TQ's time was supported by NERC grant NE/W006596/1. Abisko and Sodankylä field measurements were supported by NERC grant NE/H008187/1.





**Appendix A: Snow model intercomparisons**

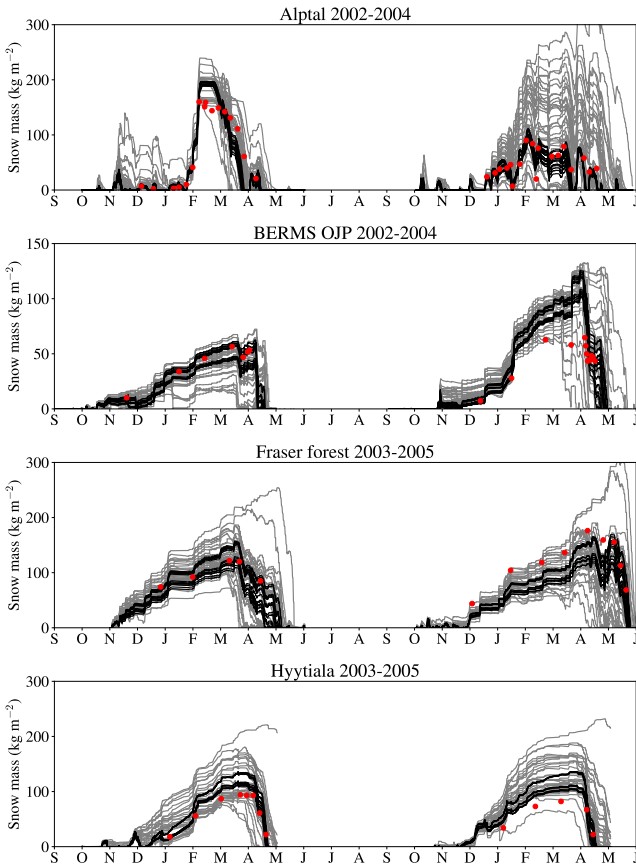

**Figure A1.** Snow mass simulated with the 16 canopy configurations of FSM2 (black lines), simulated by the 33 models that participated in SnowMIP2 (grey lines) and measured at the four SnowMIP2 forest sites (red points). Snow mass measurements for the first winter at each site were shared with the SnowMIP2 participants to allow model calibration, but the models still produced a wide range of results (Rutter et al., 2009).





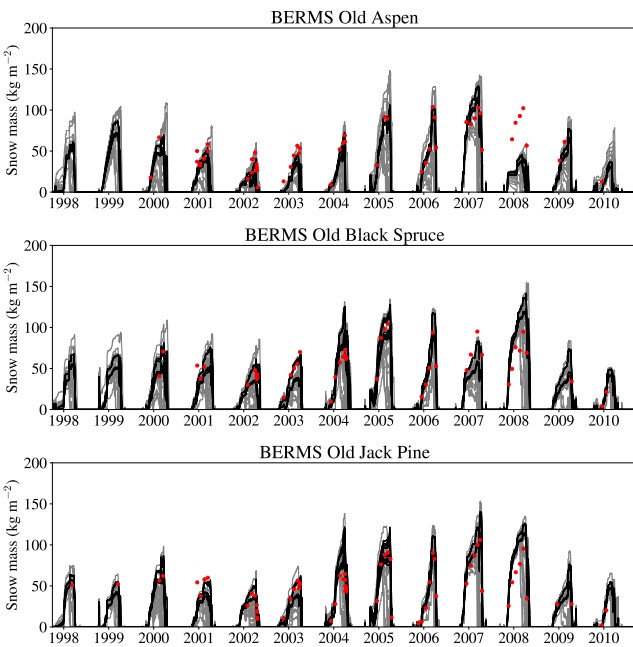

**Figure A2.** Snow mass simulated with the 16 canopy configurations of FSM2 (black lines), simulated by the 23 models that participated in ESM-SnowMIP (grey lines) and measured at the three ESM-SnowMIP forest sites (red points) (Menard et al., 2019). No data were provided for model calibration. The large underestimates in simulated snow mass at the aspen site in 2007-2008 resulted from erroneous input snowfall data.





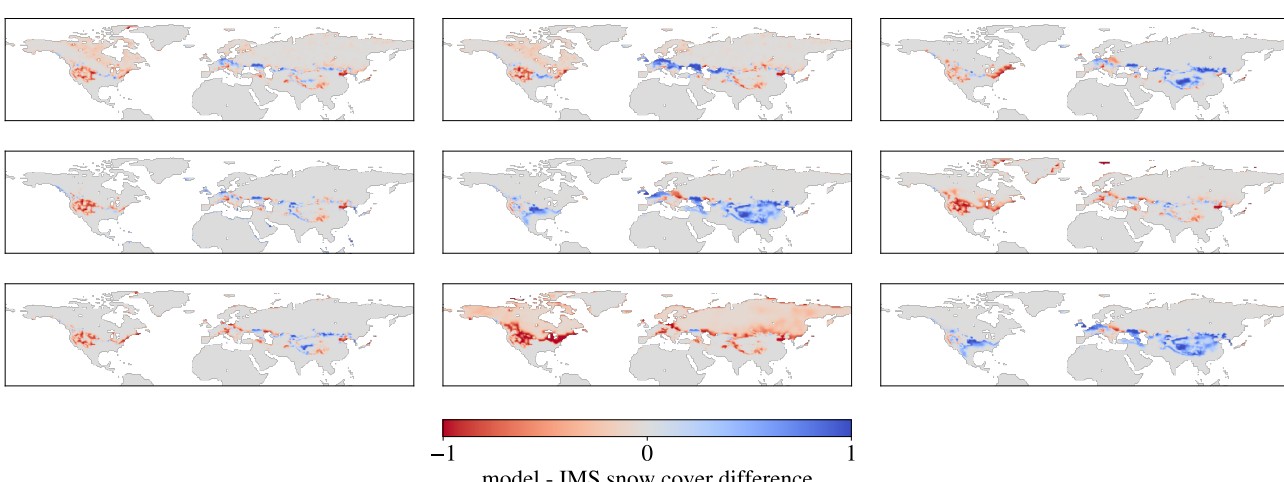

**Figure A3.** Differences between simulations of Northern Hemisphere snow cover by the nine models that participated in LS3MIP (van den Hurk et al., 2016) and IMS on 1 March 2010.



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
