# Peer review of "A Flexible Snow Model (FSM 2.1.1) including a forest canopy"

_EGUsphere, 2024_

## Author Comment (AC1)

**REVIEWER 1**

**General comments:**

- **Introduction.** In my opinion, the introduction is very solid. It is well written and guides the reader to the section describing the model. As mentioned by the authors, FSM has been widely used in previous work by many people in the snow modeling community from different countries and has contributed to the development of scientists in academia (l. 19-20). However, I think the introduction would benefit from an additional paragraph mentioning and describing the context in which researchers have used FSM in previous studies. This would highlight some shortcomings in the description of canopy snow processes in the model and underline the importance of developing a robust and flexible modelling scheme for it.

Add examples with references: "Applications have included snow data assimilation (Alonso-González et al., 2022), evaluation of snow simulations in the European Alps (Magnusson et al., 2015), Norway (Magnusson et al., 2019) and the western Himalaya (Pritchard, 2020), and construction of gridded snow datasets for the Iberian Peninsula (Alonso-González et al., 2018)".

- **Discussion.** As there are several options that have to be decided by the user, I think the authors should provide better guidance to the community on how to use their model. Based on the simulations performed at three different sites (two in Scandinavia, and one in Switzerland), based on the literature supporting the different parameterisations described in the paper, and based on their understanding of the physical processes represented in the model, I suggest that the authors provide insights regarding the options to use for specific climates or modeling purposes.

Add in the discussion "The two-layer and two-stream models in FSM2 are the more physical canopy structure and radiative transfer options, but they have relatively little influence on simulations of sub-canopy snow for the Alptal test case. The empirical canopy snow interception and unloading options make larger differences but have not been widely tested in different climates and forest types."

- **Outlook.** It would be interesting to have a few words about the ongoing and future developments of FSM2. I think the current limitations of the model should also be exposed.

Current development plans mostly relate to model couplings that extend the capability of FSM2. A limitation is that FSM2 is not a full land surface model with dynamic soil moisture.
In the Outlook, add "FSM2 is now being coupled with the Open Global Glacier Model (OGGM, Maussion et al., 2019) for physically-based mass balance modelling of glaciers; representations of firn compaction (Lundin et al., 2017) and debris cover (Reid et al., 2012) will be added for this. A limitation of FSM2 is that it does not have a dynamic representation of soil moisture; it has to be coupled with ground water and routing models for many hydrological applications".

**Specific and technical comments.**

- l. 1: I would explicitly name the model "FSM" instead of referring to a generic "model" in the first sentence of the manuscript.

Change sentence to "Multiple options for representing physical processes in forest canopies are added to FSM, which is a model with multiple options for representing physical processes in snow on the ground".

- l. 11: I feel that a sentence could be added at the end of the abstract mentioning how this new model development can help the scientific community to improve their ability to model snow in forested environments.

At the end of the abstract, add "FSM2 provides a platform for rapid investigation of sensitivity to model structure and parameter values or ensemble-based data assimilation for snow in open and forested environments".

- l. 66: Given that several models use the leaf area index (LAI) as a parameter for forest structure, and that the LAI is commonly (and rather easily) measured in the field, it is worth adding a few words on the difference between the effective vegetation index, as used in FSM, and the LAI.

Change the sentence to "models that represent transpiration or vegetation dynamics use separate leaf and stem area indices treat leaves and stems separately, but FSM2 combines them does not".
The common and easy optical methods for measuring LAI in the field are actually more directly measuring VAI.

- l. 94: Please add one or two sentences to briefly summarize the method from Erbs et al. (1982).

Add "this parametrizes the diffuse fraction as an empirical function of the ratio between global radiation at the surface and the top of the atmosphere".

- Some default parameters are mentioned in the text (l. 129, l. 256, l. 262, l. 277, l. 349) without any explanation of the choice of the specific value for these parameters. Were these values determined from a sensitivity analysis? Are they based on previous modeling or experimental work or arbitrarily chosen? Please indicate how each of these parameters was defined.

129 – add "based on Bartlett et al. (2015)"
256 – add "100 s m$^{-1}$ by default, typical of unstressed vegetation"
262 – add "by default from Dolman (1993)"
277 – add "by default from Lawrence et al. (2019)"
349 – the reference (Essery et al. 2003) is already given

- l. 171: What is the meaning of "canopy gaps"? Looking at equation 14, I understand that the authors refer to the space in a canopy layer without leaves or branches that allows the transmission of diffuse shortwave and longwave radiation. As "canopy gaps" generally refer to a sub-environment where energy and mass fluxes are altered compared to a "full canopy" environment in the snow-forest scientific literature, this could be misleading. Please consider replacing it with another term.

Change sentence to "Transmission of longwave radiation from the atmosphere through the canopy gaps".

- l. 301. Please specify the parameter from which the iteration starts.

Change sentence to "stability adjustments have to be are calculated iteratively starting from $\zeta = 0$".

- l. 353. Given the questionable physical basis of this 2/3 exponent in equation 80, I am curious about the sensitivity of the model to this parameter. Have you tried running simulations with exponents other than 2/3?

Increasing the exponent from 0 to 1 decreases the sublimation by up to 15% for Alptal simulations, but this will depend on climate, and the canopy snow cover exponent is only one of the many uncertain parameters in the parametrization options adopted for FSM2. We will have to leave a rigorous sensitivity study for future work.

- l. 365. This suggests that snow in the canopy evolves in the same way as snow on the ground. Please state this clearly in the manuscript.

Add "this is a crude approximation, but model development is limited by a lack of measurements of the evolution of intercepted snow properties (Bouchard et al. 2024)".

- l. 397 to 405: Please consider moving this to a subsection of the method in which the study sites would be described

We consider that splitting into Data and Results sections would lead to large gaps between where the measurements are described and where they are applied.

- l. 441 to 443: Please specify which parameters have been adjusted and which ones have been let as default

Change sentence to "Model parameters $\alpha_{c0}$, $\alpha_{\Lambda0}$ and $\alpha_0$ were adjusted to match measured snow-free albedos above the forest and the meadow".

- Figure 8: Some plots are somehow a bit messy with all the black lines (especially 8b). Consider showing the ensemble median as a black line with a min-max envelope.

Although less "messy", a min-max envelope conceals whether the simulations are evenly distributed or clustered. Here is what the figure suggested by the reviewer would look like, but we prefer the original:

[Figure]

- l. 446-447. I imagine that a subjective discrimination between 9 levels of canopy interception must be very difficult to do. Showing photos of these 9 interception levels would help the reader to see the nuance between each stage of canopy load. I suggest adding this in a supplementary material document.

This surely is difficult, but we did not make these observations and do not have the photos that the reviewer asks for; the data are from Stähli et al. (2009). Emphasize this by adding "Figure 8 compares simulations with observations from Stähli et al. (2009) of …".

- Table 1: This is perhaps a personal preference, but as a reader I would find it easier to grasp differences in modelling results between each simulation with a figure rather than a table. This could be done by plotting two variables against each other on an x-y plot and a color map for the third variable. As there are only 16 points to plot, you could then write a short name for each simulation on the graph without oversaturating it.

This is a nice idea that demonstrates the correlations mentioned in the text. A colour scale cannot show ranking of the third variable as precisely as a table and the labels have to be explained in a table, so Table 1 is still required alongside this new Figure 9:

[Figure]

**Figure 9.** Maximum sub-canopy snow mass, duration of snow cover on the ground and fraction of total snowfall sublimating in the 16 Alptal forest simulations numbered in Table 1.

- l. 521. Can you explain why this particular model behaves differently to the others?

We suspect that this is because of differences in how this model parametrizes snow cover fraction in its surface energy balance calculations, but this paper is not about the performance of the LS3MIP models.

- I find that Figure 9b is very interesting. I would like to see how this partitioning would change with the different simulation schemes (Table 1). I am curious about what drives the melt energy for sites with different canopy densities. Consider including this in the supplementary material.

Modify Figure 9b to show partitioning for all simulation schemes and dominant components driving melt energy for different canopy densities:

[Figure]

- Figure 10. Please enlarge this figure.

This figure has been enlarged in the manuscript. Its size in a final paper will depend on the journal.

- l. 554-555. Can you elaborate on how these modeling choices regarding canopy snow unloading and canopy snow-rainfall interactions may be more appropriate for some climates and less appropriate for others?

Add "Rain falling on snow is a rare phenomenon occurring most frequently in the Arctic and northern maritime regions that can have large impacts (Cohen et al., 2015)".

- l. 571-572: Please, revise this sentence. There seems to be a word missing.

With apologies, we cannot see an error in this sentence: "Different canopy radiative transfer parametrizations, however, need not result in large differences in masking of snow albedo by forests."

**REVIEWER 2 / ISABELLE GOUTTEVIN**

- Dense vs sparse canopies are sometimes mentioned, with impacts on the model parameters (e.g. L 257; L 285-287). It is not very clear by what is meant with these words (occurrence of canopy gaps or homogeneous, sparse canopies, or both?) and maybe, should be made more explicit. How is it done in large-scale simulations with explicit forest fractions? (L513-514)

For FSM2, "dense" canopies are the limit $\Lambda \to \infty$ and "sparse" canopies are ones that have $\Lambda$ less than the asymptotic limit above which simulations are independent of canopy density. Each grid cell in the large-scale simulations has a forest fraction, which has a canopy density.
Change line 125 to "dense canopy albedo $\alpha_c$ in the limit $\Lambda \to \infty$".
Change line 493 to "By design in FSM2, simulations are continuous as $\Lambda \to 0$ for sparse canopies and are independent of canopy density for dense canopies with $\Lambda \gtrsim 4$".
Change line 515 to "simulations with parameters for evergreen forest, deciduous forest and unforested land are combined according to their fractions".

- L 95 / section 2.2 or 2.2.4: I missed some background, here or in the introduction, on the Beer's Law vs Two-stream approximation options. What would motivate one over the other in general, and in forest-specific literature?

At the end of 2.2, add "Beer's Law and the two-stream approximation can give similar predictions of broadband canopy albedo and transmission for surface energy balance calculations (Essery, 2013), but the two-stream approximation is more accurate for spectrally-resolved calculations (Wang, 2003)".

- L 129 / eq 15, the choice for the default values of the snow-free and snow-covered dense canopy albedo parameters should be justified

See response to Reviewer 1 on choice of parameter values. The default value for snow-covered canopy albedo $\alpha_{cs} = 0.4$ is a little higher than typical literature values, so results have been recalculated with $\alpha_{cs} = 0.3$ (the value used in Gouttevin et al., 2015). The changes in results are small.

- L 139: wouldn't there be a typo here, and wouldn't a_Lambda rather be alpha_Lambda?

Corrected equation "$\omega = \alpha_\Lambda \cancel{a_\Lambda}$".

- Fig 3 and 5: Has such a comparison between the two kinds of radiative transfer models for canopy ever been produced, and with similar outcomes? I find these very interesting results.

Essery (2013) and Qu and Hall (2014), already cited, are examples of such comparisons with contrasting results. See addition to section 2.2, above, also.

- L 312: a comma after "fluxes" may clarify the sentence

We have tried this; making it a parenthetical clause also requires adding a comma on the end of equation 70, which does not look clearer.

- L 345: It seems to me that VAI has not been defined earlier in the text; it should be defined line 66.

Change line 66 to "effective vegetation area index (VAI) $\Lambda$".

- L 372: It seems to me that Sf has not been introduced; it could probably be done L 370.

Add "for snow falling at rate $S_f$" to the end of the sentence starting in line 370.

- L 374: If I am correct, Eq 84 holds for each layer with its respective VAI to drive Sc; I feel it would be worth mentioning it to make it easier to understand the effects on interception when going to the 2-layer version.

Add "Eqs 84 or 85 are applied in both layers of the two-layer canopy model, with interception in the upper layer subtracted from snowfall reaching the lower layer".

- L 460-461: it would be worth mentioning that SnowMIP2 simulations involved models with a very crude, parametric representation of canopies, which probably explains a larger scatter in these simulations than in the FSM2-ensemble.

Add "Essery et al. (2009) suggested that this spread was largely due to uncertain parameter selections for highly simplified canopy models".

- L 483-484: I remember getting on average higher subcanopy longwave radiations upon the use of a 2-layer canopy model in SNOWPACK, despite indeed lower daily values (see Table 3 in Gouttevin

et al. 2015). There are several differences between the 2-layer model you developed and the 2-layer canopy model of SNOWPACK but I'm curious if you have an explanation for this?

With many differences in structure and parameter values, explaining differences would be very difficult without a detailed study running both models side by side. This is a difference worth noting, however. In the discussion, add "Using two-layer canopy options decreases the diurnal ranges of sub-canopy downward longwave radiation in both FSM2 and SNOWPACK. In SNOWPACK, Gouttevin et al. (2015) show in their Fig. 3 show that this reduction is dominated by an increase in nighttime minima and daily averages are increased. Heating of the lower layer in clear days and cooling in clear nights are both reduced in FSM2, and difference in daily averages are small (Fig. 8 here)."

- Figure 11: it would be nice to mention an explanation for the snow mass overestimations by most models, if there exists one.

Add "Mudryk et al. (2020) noted that the observational estimates of hemispheric snow mass are likely to be biased low because of underestimation in mountain regions".